# The patient, diagnostic, and treatment intervals in adult patients with cancer from high- and lower-income countries: A systematic review and meta-analysis

Dafina Petrova[1,2,3]*, Zuzana Špacírová[1,2,3], Nicolás Francisco Fernández-Martínez[1,2,3,4], Ana Ching-López[1,2,3], Dunia Garrido[5], Miguel Rodríguez-Barranco[1,2,3], Marina Pollán[3,6], Daniel Redondo-Sánchez[1,2,3], Carolina Espina[7], Camila Higueras-Callejón[2], Maria José Sánchez[1,2,3,8]

1 Instituto de Investigación Biosanitaria ibs.GRANADA, Granada, Spain, 2 Escuela Andaluza de Salud Pública (EASP), Granada, Spain, 3 CIBER of Epidemiology and Public Health (CIBERESP), Madrid, Spain, 4 Interlevel Clinical Management Unit for Prevention, Promotion and Health Surveillance, Reina Sofía University Hospital, Córdoba, Spain, 5 Department of Developmental and Educational Psychology, University of Granada, Granada, Spain, 6 National Center for Epidemiology, Health Institute Carlos III, Madrid, Spain, 7 International Agency for Research on Cancer (IARC/WHO), Lyon, France, 8 Department of Preventive Medicine and Public Health, University of Granada, Granada, Spain

* dafina.petrova.easp@juntadeandalucia.es

**Data Availability Statement:** All supporting data and analysis for the review can be downloaded

## Abstract

### Background

Longer time intervals to diagnosis and treatment are associated with worse survival for various types of cancer. The patient, diagnostic, and treatment intervals are considered core indicators for early diagnosis and treatment. This review estimated the median duration of these intervals for various types of cancer and compared it across high- and lower-income countries.

### Methods and findings

We conducted a systematic review with meta-analysis (prospectively registered protocol CRD42020200752). Three databases (MEDLINE, Embase, and Web of Science) and information sources including grey literature (Google Scholar, OpenGrey, EThOS, ProQuest Dissertations & Theses) were searched. Eligible articles were published during 2009 to 2022 and reported the duration of the following intervals in adult patients diagnosed with primary symptomatic cancer: patient interval (from the onset of symptoms to first presentation to a healthcare professional), diagnostic interval (from first presentation to diagnosis), and treatment interval (from diagnosis to treatment start). Interval duration was recorded in days and study medians were combined in a pooled estimate with 95% confidence intervals (CIs). The methodological quality of studies was assessed using the Aarhus checklist.

A total of 410 articles representing 68 countries and reporting on 5,537,594 patients were included. The majority of articles reported data from high-income countries ($n = 294$, 72%),

from the Open Science Framework: DOI 10.17605/OSF.IO/REY9C (https://osf.io/rey9c/).

**Funding:** This work was supported by the Spanish Association against Cancer (Asociación Española contra el Cáncer, PROYE20023SÁNC "High resolution study of social inequalities in cancer (HiReSIC)" to MJS), the Cancer Epidemiological Surveillance Subprogram of the CIBER of Epidemiology and Public Health and the Health Institute Carlos III (VICA to MJS), and the Health Institute Carlos III (PI18/01593 "Multilevel population-based study of socioeconomic inequalities in the geographical distribution of cancer incidence, mortality and net survival" to DP). DP is supported by a Juan de la Cierva Fellowship from the Ministry of Science and the National Research Agency of Spain (MCIN/AEI, JC2019-039691-I, http://doi.org/10.13039/501100011033, Accessed 4 October 2021). The funders had no role in study design, data collection and analysis, decision to publish, or preparation of the manuscript.

**Competing interests:** The authors have declared that no competing interests exist.

**Abbreviations:** AIDS, acquired immunodeficiency syndrome; CI, confidence interval; GNI, gross national income; HDI, Human Development Index; ICP, Index of Cancer Preparedness; NCDB, National Cancer Database; SD, standard deviation; SEER, Surveillance, Epidemiology, and End Results; WOS, Web of Science.

with 116 (28%) reporting data from lower-income countries. Pooled meta-analytic estimates were possible for 38 types of cancer. The majority of studies were conducted on patients with breast, lung, colorectal, and head and neck cancer. In studies from high-income countries, pooled median patient intervals generally did not exceed a month for most cancers. However, in studies from lower-income countries, patient intervals were consistently 1.5 to 4 times longer for almost all cancer sites. The majority of data on the diagnostic and treatment intervals came from high-income countries. Across both high- and lower-income countries, the longest diagnostic intervals were observed for hematological (71 days [95% CI 52 to 85], e.g., myelomas (83 days [47 to 145])), genitourinary (58 days [50 to 77], e.g., prostate (85 days [57 to 112])), and digestive/gastrointestinal (57 days [45 to 67], e.g., colorectal (63 days [48 to 78])) cancers. Similarly, the longest treatment intervals were observed for genitourinary (57 days [45 to 66], e.g., prostate (75 days [61 to 87])) and gynecological (46 days [38 to 54], e.g., cervical (69 days [45 to 108]) cancers. In studies from high-income countries, the implementation of cancer-directed policies was associated with shorter patient and diagnostic intervals for several cancers.

This review included a large number of studies conducted worldwide but is limited by survivor bias and the inherent complexity and many possible biases in the measurement of time points and intervals in the cancer treatment pathway. In addition, the subintervals that compose the diagnostic interval (e.g., primary care interval, referral to diagnosis interval) were not considered.

## Conclusions

These results identify the cancers where diagnosis and treatment initiation may take the longest and reveal the extent of global disparities in early diagnosis and treatment. Efforts should be made to reduce help-seeking times for cancer symptoms in lower-income countries. Estimates for the diagnostic and treatment intervals came mostly from high-income countries that have powerful health information systems in place to record such information.

---

### Author summary

#### Why was this study done?

- Cancer is a leading cause of death globally and timely diagnosis and treatment are considered essential for improving cancer outcomes.

- Three main intervals describe the time patients spend in the pathway to treatment of cancer: the patient interval (from symptom start to first presentation to a healthcare professional), the diagnostic interval (from first presentation to diagnosis), and the treatment interval (from diagnosis to the start of treatment).

- The duration of these intervals could vary greatly depending on the type of cancer and the socioeconomic level of the country.

## What did the researchers do and find?

- We conducted a systematic review with meta-analysis of the duration of the patient, diagnostic, and treatment intervals in adult patients with diverse types of cancer.

- We included 410 articles representing 68 countries and reporting on 5,537,594 patients; the majority of articles reported data from high-income countries (72%), with only 28% reporting data from lower-income countries.

- Patient intervals in studies from lower-income countries were consistently 1.5 to 4 times longer that patient intervals from studies from high-income countries for almost all cancer sites. The majority of data on the diagnostic and treatment intervals came from high-income countries, and there was large variation according to the type of cancer.

## What do these findings mean?

- These results identify the cancers where diagnosis and treatment initiation may take the longest and reveal important global disparities in early diagnosis and treatment.

- Efforts should be made to reduce help-seeking times for cancer symptoms in lower-income countries and conduct more research in lower-income contexts, especially on the intervals to diagnosis and treatment.

- This review summarized a large number of studies conducted worldwide but is limited by biases that could arise due to patient selection (e.g., only patients who survived a certain amount of time) and the difficulty of accurately measuring time intervals for past events.

Cancer is a leading cause of death globally, accounting for nearly 10 million deaths worldwide in 2020 [1]. Timely diagnosis and treatment are considered essential for improving cancer outcomes [2]. In its guide to early cancer diagnosis, the World Health Organization considers stage at the time of diagnosis and the duration of the patient, diagnostic, and treatment intervals core indicators for early diagnosis and treatment [2]. These intervals are defined in the Model of Pathways to Treatment [3,4] and together describe the entire duration of time spent in the pathway to treatment of symptomatic cancer in a way applicable to most, if not all, healthcare systems and cancer types. The patient interval describes the time from symptom start to first presentation (i.e., the first presentation to a healthcare professional). The diagnostic interval represents the time elapsed between first presentation and diagnosis, and the treatment interval the time from diagnosis to the start of treatment [4]. The duration of these intervals is likely a combination of time that is both necessary or unavoidable (e.g., need for additional diagnostic workup; need for patients to recover and become physically fit to undergo treatment) and time that is avoidable and should be reduced (e.g., presentation delays due to ignorance or fear from cancer; scheduling delays due to an overburdened healthcare system).

It is generally expected that longer interval duration is associated with worse cancer outcomes such as later stage at diagnosis and higher mortality [2,5]. Consistent with this, there is

evidence to suggest that shorter times to diagnosis are associated with better outcomes in terms of stage at diagnosis and survival for breast, colorectal, head and neck, testicular cancers, and skin melanoma, with less evidence for pancreatic, prostate, and bladder cancers [6–10]. Longer treatment intervals, even only a 4-week delay in surgery, systemic treatment, and/or radiotherapy, are also associated with higher mortality for 7 cancers including bladder, breast, colorectal, cervical, and head and neck cancers [11–15]. There is likely to be large variation between cancers in terms of the benefit (or lack thereof) of shorter intervals. However, besides "hard" oncological outcomes, we should consider that patients may generally appreciate and benefit from timely diagnostic and treatment workup on other outcomes such as anxiety, emotional distress, and quality of life [16].

The majority of the evidence on the effects of early cancer diagnosis and treatment on patient outcomes comes from high-income countries [6,8,10–15]. It is thus not clear to what extent waiting time thresholds established in higher-income contexts (e.g., for referral to specialist care or initiation of treatment) would have similar effects or be equally feasible in lower-income countries [1,17,18]. For example, when it comes to the duration of intervals on the cancer care pathway, previous research indicates that these vary greatly depending on not only the type of cancer diagnosed [19] but also the socioeconomic level of the country [20]. To illustrate, low-income countries are characterized by significantly longer patient intervals than middle-income countries [20]. Common barriers to early diagnosis and care such as poor health literacy, cancer stigma, lack of access to diagnostic tests and treatment services, and financial, geographical, or logistical barriers are likely to be exacerbated in lower-income contexts, contributing to longer intervals on the cancer care pathway and worse patient outcomes [2,20,21]. Longer times to diagnosis and treatment and later stage at diagnosis are likely some of the multifactorial patient- and health system-driven causes of the larger cancer burden and lower survivorship in lower-income countries [22].

Previous reviews have offered information about the duration of different intervals focusing on specific cancers [23–26]. Another recent review reported on the duration of different intervals for childhood and breast cancer in lower-income countries [20]. However, there has been no review that offers an overview of the duration of the different intervals across different cancer sites and comparing high- and lower-income countries. Until recently, there was also no validated methodology for reliably combining median interval duration data using meta-analytic techniques [27]. Hence, the goal of the current research was to conduct a systematic review with meta-analysis of the duration of the patient, diagnostic, and treatment intervals in adult patients with diverse types of cancer and to compare this duration between high- and lower-income countries.

## Method

We followed PRISMA 2020 guidelines in conducting and reporting the meta-analysis [28]. The review protocol was prospectively registered in PROSPERO with ID CRD42020200752.

### Literature search

Following published recommendations for optimal database selection [29] and in close collaboration with the first author, a librarian designed and implemented a search strategy in MEDLINE (via Ovid), Embase, and Web of Science (WOS)-Core Collection. The strategy was initially designed for MEDLINE (Ovid), which combined MeSH terms and keywords, and subsequently adapted for the rest of bibliographic databases including the use of EMTREE controlled vocabulary in Embase database. Other sources of information were also explored to identify grey literature (Google Scholar, OpenGrey, EThOS, and ProQuest Dissertations &

Theses). The full search strategy, informed by the PRISMA-S extension [30], is available in S1 Text. The period searched was initially from January 1, 2009 to September 1, 2020 and was then updated until May 19, 2022, following initial peer review. The starting date was chosen based on (a) the date of publication of the Olesen Model [31] and the Model of Pathways to Treatment [3], 2 seminal publications about the different intervals on the cancer care pathway; and (b) with the purpose to include only fairly recent evidence. There were no restrictions by language or country. Additional studies were identified by reviewing the reference lists of relevant studies identified from the search.

## Inclusion criteria

Studies reporting data on the length of any of the 3 intervals of interest for any cancer site in adult patients with cancer presenting with primary cancers were included. The intervals were defined according to the Aarhus statement [4]. The patient interval was defined as time from the date of first symptom to the date of first presentation, i.e., first contact with a healthcare professional. The diagnostic interval was defined as time from the date of first contact with a healthcare professional to the date of diagnosis. Finally, the treatment interval was defined as time from the date of diagnosis to the date of start of the first treatment. In the case of the patient and diagnostic intervals, only studies of symptomatic patients were considered (i.e., excluding screening or accidentally detected cancers). As a minimum, studies had to report the median or mean duration of the interval in days (weeks and months were converted to days, multiplying by 7 and 30, respectively) and the number of patients.

## Exclusion criteria

Studies not reporting the results of original work, qualitative studies not reporting interval duration, studies reporting mostly on patients diagnosed with asymptomatic cancers (i.e., through screening), studies reporting mostly on patients with secondary/relapse cancer, studies reporting on children, adolescents, and/or young adults (defined as mean sample age <30 years), studies not reporting intervals for specific cancer sites, and studies reporting hypothetical intervals (e.g., help-seeking intervals from surveys with healthy populations) were excluded. If studies reported intervals for periods after the start of the coronavirus pandemic, those were excluded retaining only intervals prior to the pandemic. Systematic reviews and meta-analyses were excluded, but, if relevant, their reference lists were manually searched to identify further original studies.

## Article selection

The Covidence software (https://www.covidence.org) was used for the systematic review management. Because we expected to identify a large number of abstracts for screening, to reduce reviewer workload, we planned to perform the screening individually (i.e., that abstracts be screened by 1 reviewer only), if we could establish that agreement between reviewers was sufficiently high. To assess this, we performed independent and blind screening of 26% of the abstracts by 2 reviewers. Agreement was satisfactory against the preestablished criterion of >90% (i.e., agreement for the 10 pairs of reviewers varied between 87% and 100%), and, after discussion of the disagreements, screening was continued individually.

The full text of selected studies was independently screened against the inclusion/exclusion criteria by 2 reviewers blinded to each other's decisions. Disagreements were documented and resolved by discussion or a third reviewer. An exception was made for articles considered after the literature search update, where the first author acted as an arbiter in case of disagreement. Reasons for exclusion were documented.

## Data extraction

This was performed in the Covidence tool (study and population characteristics) and in a spreadsheet (statistical results) by 2 reviewers. Disagreements were resolved through discussion or a third reviewer (except for articles considered after the literature search update, where the first author acted as an arbiter in case of disagreement). For each study, we recorded year of publication, country, total number of patients, study setting, data sources, study design, inclusion and exclusion criteria, cancer site, type of interval studied, and participant characteristics. For each interval, the following statistical information was recorded if available (in days): median, interquartile range, minimum, maximum, mean, standard deviation (SD), sample size (N), country, year of start and end of data collection (data were recorded separately for different years if reported per year), cancer site, specific diagnosis, mode of diagnosis confirmation, and type of first treatment if specified (relative to the treatment interval). Because many studies used the same large databases, after data extraction was completed, the first author revised the resulting dataset to perform additional control for duplicate samples. When 2 studies reported interval data for the same cancer site and based on largely the same population, the study with larger sample size and/or more inclusive criteria was retained.

## Country socioeconomic indicators

To separate countries into high- and lower-income economies, 2 socioeconomic country indicators were extracted for each study by an expert health economist: the gross national income (GNI) and the Human Development Index (HDI) (see S2 Text for details). Following a previous meta-analysis [32], the indicators were extracted for each study according to the respective country and year in which data were collected, to represent the country's development during the time of diagnosis and treatment.

In addition, to further explore variability within high-income countries only, we extracted the Index of Cancer Preparedness (ICP): Policy and Planning [33]. This index offers a quantitative measure of the quality of policies aimed to control cancer based on multiple indicators such as the existence and comprehensiveness of a national cancer plan, cancer registries, policies regarding tobacco control, lifestyle and diet, and cancer research, among others (see S2 Text for details).

## Risk of bias

This was evaluated using a short form of the "Aarhus checklist" [4] developed to assess the quality of studies that measure intervals on the cancer treatment pathway. The checklist contains questions regarding interval definitions, measurement, use of theoretical frameworks, discussion of validity, biases, and limitations of measurement, among others. The checklist was completed independently by 2 reviewers, and disagreements were resolved by a third reviewer. Studies with scores <25% were considered high risk and studies with ≥75% low risk, with the rest considered intermediate (see S3 Text).

## Statistical analysis

As expected, the most often reported statistic for the duration of the intervals was the median, and meta-analysis was conducted with the "metamedian" package (v.0.1.5) in R (v.4.1.1) and following McGrath and colleagues [27]. Specific study medians (or means in a minority of occasions, when medians were not reported, as per McGrath and colleagues [27]) were combined in a pooled median, and 95% confidence intervals (CIs) were calculated [27]. The meta-analytical methods available for medians do not provide an estimate of heterogeneity;

however, we used the "median of medians" method, which is more suitable for heterogenous data [27].

Specific cancer sites were further grouped following the categorization of the National Cancer Institute [34] into the following main cancer groups: acquired immunodeficiency syndrome (AIDS)-related, breast, digestive/gastrointestinal, endocrine/neuroendocrine, genitourinary, gynecologic, head and neck, hematologic/blood, musculoskeletal, neurologic, respiratory/thoracic, skin, and unknown primary.

To investigate to what extent pooled medians were different as a function of country level indicators, several approaches were used. First, stratified meta-analyses were performed for high- versus lower-income countries (high versus lower GNI and higher versus lower HDI). In the case of HDI, the higher versus lower groups were created using k-means clustering (i.e., the groups were based on their "natural" grouping based on k = 2 centroids). This method was chosen because of the skewed distribution of the HDI variable, which would result in artificial grouping using other methods such as creating equal-count groups. Second, differences in interval duration (in number of days) between studies conducted in high- and lower-income countries were estimated based on Wilcoxon rank sum tests, generating a 95% CI for the estimated differences. Third, these analyses were complemented with a random-effects meta-regression analysis in the "metafor" (v.3.0.2) package in R [35]. In this analysis, the study-specific medians were declared as a "GEN" measure, and studies were weighed analogous to weighting in the "metamedian" package (proportional to the number of subjects and normalized). The GNI group (high versus lower) and the HDI score (continuous and centered at the mean) were individually tested as moderators. We extracted p-values for the moderator tests and the percentage of variance explained by the moderator ($R^2$). To further explore variability within high-income countries only, we conducted analogous analyses using the ICP: Policy and Planning index only considering studies conducted in high-income countries (based on GNI).

To compare and estimate the relative contribution of the different intervals, following previous studies [19], we calculated the ratios between the different intervals. Specifically, because the diagnostic interval was the longest interval for the majority of cancers, following methods by Bonett and Price [36], we calculated the ratios of the diagnostic to the patient interval (DI/ PI), the diagnostic to the treatment interval (DI/TI), and the patient to the treatment interval (PI/TI) with their respective 95% CI. This was only done for studies that reported the duration of all 3 intervals in the same sample of patients and intervals were considered to be significantly different when the 95% CI for their ratio excluded 1.

Sensitivity analysis included repeating the main analysis after excluding studies with high risk of bias according to the Aarhus checklist and after excluding studies that did not report the median and the mean was therefore imputed as median (even though the "metamedian" package can reliably estimate a pooled median when the mean is reported instead of the median for a small proportion of studies, in our case, this was n = 52 (14%), n = 15 (5%), and n = 70 (15%) for the patient, diagnostic, and treatment interval, respectively). However, using means as medians can introduce bias when means are not a good approximation of the medians (i.e., due to a skewed distribution), and, hence, we wanted to investigate if the inclusion of means introduced such bias in the analyses.

## Results

Initially, 12,140 records were retrieved and 410 articles were finally included in the review. Fig 1 shows a detailed flow chart of the review selection process. All studies excluded at the full text stage are listed in S1 Table along with individual reasons for exclusion. Full bibliographic details of the included studies are available in S2 Table.

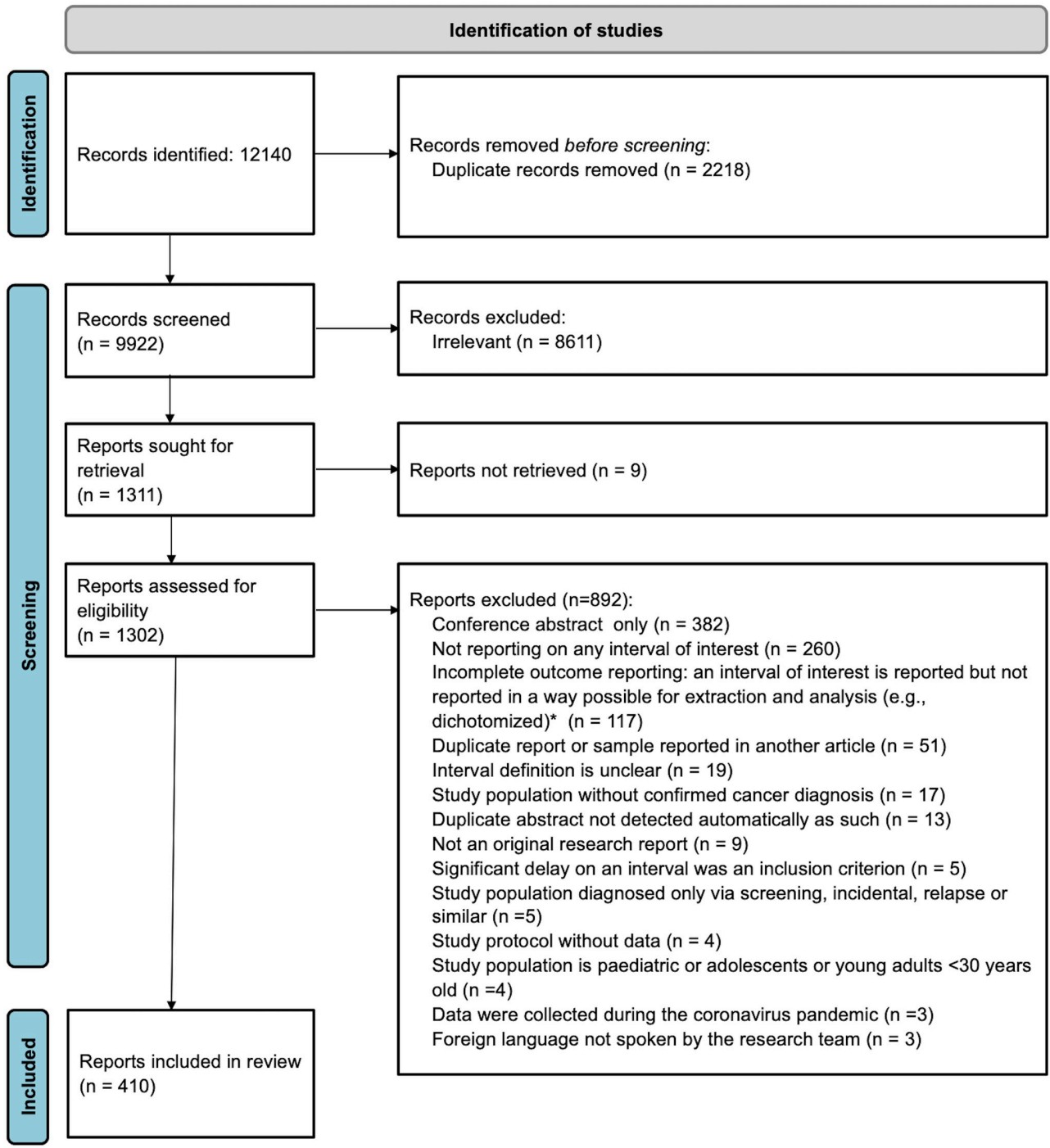

**Fig 1. PRISMA flow chart of the study selection process.** *Missing information was not requested from authors.

The articles reported data from 68 countries (see Fig 2 and S3 Table), the most frequently represented being the United States (22%), United Kingdom (8%), Canada (7%), and the Netherlands, Denmark, Australia, and Spain (4% each). The majority of articles reported data from high-income countries (294 or 72%), with 116 (28%) reporting on lower-income countries: 48 (12%) reporting data on upper-middle, 43 (10%) on lower-middle, and 24 (6%) on

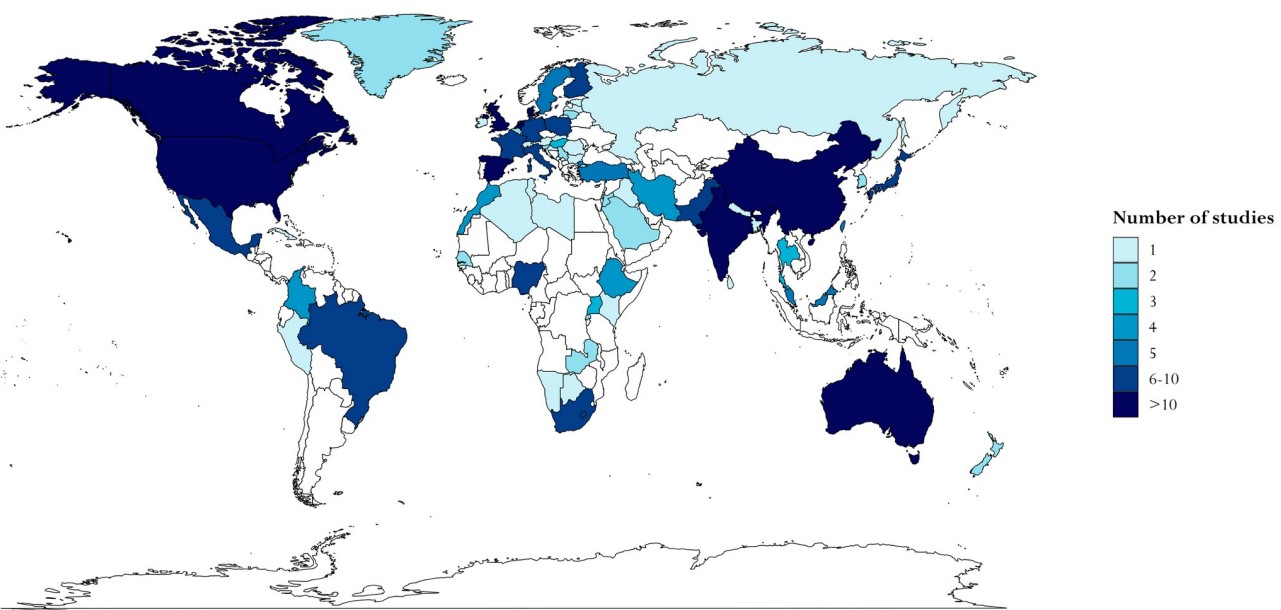

**Fig 2. Choropleth map of countries represented in the systematic review according to the number of studies in which each country is represented.**
The base layer map is obtained in R via ggplot2::map_data("world"), which imports the world map from Natural Earth, which is in the public domain and available from https://www.naturalearthdata.com, with terms of use available in http://www.naturalearthdata.com/about/terms-of-use/.

low-income countries. The majority of studies (88%) were cross-sectional studies. S4 Table contains detailed information about each included study, including specific inclusion and exclusion criteria, study setting, number of participants, design, and data sources.

The studies included a total of 5,537,594 patients: 85,609 with data on the patient, 266,331 on the diagnostic, and 5,166,938 on the treatment interval, respectively. Of the 410 articles included, 103 (25%) reported on more than one of the intervals of interest. Regarding the individual intervals, 177 (43%) reported data on the patient, 121 (30%) on the diagnostic, and 243 (59%) on the treatment interval. Reporting the patient interval was more common in studies from lower-income countries ($n = 89$, 77% versus $n = 88$, 30% of studies from high-income countries), whereas reporting the treatment interval was more common in studies from high-income countries ($n = 194$, 66% versus $n = 49$, 42% of studies in lower-income countries). The percentage of studies reporting the diagnostic interval was similar for high- and lower-income countries, with $n = 86$ (29%) versus $n = 35$ (30%), respectively.

One hundred (24%) articles used a combination of sources to obtain interval data. In particular, 146 (36%) used questionnaires or interviews (with patients or health professionals), 215 (52%) used medical records, and 141 (34%) used large databases (e.g., population-based cancer registry data, the USA National Cancer Database (NCDB), Surveillance, Epidemiology, and End Results (SEER), or similar). Use of questionnaires or interviews was more frequent in studies from lower-income countries ($n = 77$, 66% versus $n = 69$, 23% of studies from high-income countries), whereas use of large databases was more frequent in studies from high-income countries ($n = 137$, 47% versus $n = 4$, 3% of studies from lower-income countries). The use of medical records was relatively similar ($n = 150$, 51% for high-income, and $n = 65$, 56% for studies from lower-income countries, respectively).

The mean risk of bias score for the sample was 51% (SD = 22%). Fifty-three studies (13%) received a high, 290 (71%) medium, and 67 (16%) low risk of bias score on the Aarhus checklist. Among the items that applied to all studies, the following results should be noted: Only

19% of studies made a reference to a theoretical framework (or the need for one); 72% provided interval definitions that the review team judged as precise, transparent, and reproducible; and 68% fully described the healthcare context of the study. The individual ratings for each study are available in S5 Table.

## Meta-analyses results

In the case of AIDS-related and endocrine/neuroendocrine tumors, only 1 study for each group was identified, and, hence, these main groups are not further discussed. All studies reporting on the respiratory/thoracic group were lung cancer studies. Tables 1, 2 and 3 report the results of the meta-analyses of the patient, diagnostic, and treatment intervals, respectively, for each main cancer group and specific site for which studies were identified, and as a function of the socioeconomic level of the country. Overall, meta-analyses for 38 cancer sites could be conducted.

**Patient intervals.** There were significant and very pronounced differences on the patient interval between high- and lower-income countries for almost all cancer groups explaining between 7% and 55% of the variance in the study-specific medians (with the exception of genitourinary cancers for which there were only k = 2 studies from lower-income countries; see S6 Table for statistical results). For this reason, in the following, we consider results for high- and lower-income countries separately.

In studies from high-income countries (high GNI), median patient intervals did not exceed a month for the majority of cancer sites (i.e., the 75th percentile of the distribution of pooled medians was 31 days), suggesting that, in these countries, at least half of patients generally see a healthcare professional within a month from symptom onset. Among the main cancer groups, the longest pooled patient intervals were observed for skin cancers (melanoma: 70 days [95% CI 25 to 764] and nonmelanoma: 60 days [57 to 477]), breast (32 days [24 to 45]), and head and neck (31 days [30 to 58]) cancers (see Table 1). Studies that focused exclusively on pregnancy-associated breast cancer reported a significantly longer patient interval (48 days [30 to 61]) compared to general (nonpregnancy-associated) breast cancer studies (29 days [19 to 45]). For the rest of the main cancer groups, the median patient interval varied between 18 and 26 days (digestive/gastrointestinal: 26 days [21 to 30]; lung: 22 days [20 to 30]; gynecological: 21 days [15 to 30]; hematological: 19 days [16 to 26], and genitourinary: 18 days [10 to 30]).

Within lower-income countries, median patient intervals generally exceeded 1.5 months for the majority of cancer sites (i.e., the 25th percentile of the distribution of pooled medians was 53 days). Among the main cancer groups, the longest pooled patient intervals were observed for neurologic/brain cancers (270 days [48 to 730]), skin melanoma (118 days [85 to 150]), genitourinary (97 days [78 to 116]), and gynecological cancers (79 days [40 to 129]), followed by head and neck (60 days [45 to 90]), breast (58 days [35 to 92]), digestive/gastrointestinal (53 days [34 to 125]), hematological (42 days [23 to 156]), and lung (33 days [28 to 59]) cancers (see Fig 3).

Overall, patient intervals from lower-income countries were about 1.5 to 4 times longer than those found in studies from high-income countries and varied generally between 1 and 4 months (see Table 1). The pooled medians were significantly longer for lower-income countries for all main cancer groups with the exception of genitourinary and skin cancers, where the number of studies from lower-income countries was very small (2 for each group) (see Table 1 and Fig 3). Notable differences between lower- and high-income countries included a pooled median of 90 days [43 to 279] compared to 29 days [22 to 32] for colorectal cancer ($R^2$ = 25% based on both GNI and HDI); 79 days [40 to 129] versus 21 days [15 to 30] for

Table 1. Meta-analysis results from studies reporting on the patient interval (PI).

| Body location/system | Cancer site | PI All studies | | | | | PI Studies from high-income countries | | | | | PI Studies from lower-income countries | | | | | PI Difference | | | |
|---|---|---|---|---|---|---|---|---|---|---|---|---|---|---|---|---|---|---|---|---|
| | | k | N | med | LLCI | ULCI | k | N | med | LLCI | ULCI | k | N | med | LLCI | ULCI | Diff | LLCI | ULCI | p |
| **Breast** | All breast* | 93 | 24,957 | 43 | 34 | 59 | 26 | 6,911 | 32 | 24 | 45 | 67 | 18,046 | 58 | 35 | 92 | 25 | 7 | 61 | 0.002 |
| | Breast not PA-associated | 87 | 24,812 | 42 | 34 | 55 | 22 | 6,789 | 29 | 19 | 45 | 65 | 18,014 | 50 | 34 | 90 | 24 | 8 | 61 | 0.002 |
| | Breast PA-associated | 6 | 145 | 61 | 32 | 195 | 4 | 113 | 48 | 30 | 61 | 2 | 32 | 196 | 109 | 282 | 144 | 48 | 252 | 0.100 |
| **Digestive/gastrointestinal** | All digestive/gastrointestinal | 50 | 12,729 | 29 | 22 | 31 | 42 | 11,612 | 26 | 21 | 30 | 8 | 1,117 | 53 | 34 | 125 | 27 | 10 | 69 | 0.002 |
| | All upper gastrointestinal** | 17 | 2,678 | 15 | 11 | 28 | 16 | 2,554 | 15 | 10 | 23 | 1 | 124 | 33 | | | 18 | −46 | 29 | 0.260 |
| | Anal | 0 | | | | | 0 | | | | | 0 | | | | | | | | |
| | Colon | 2 | 1,747 | 20 | 18 | 22 | 2 | 1,747 | 20 | 18 | 22 | 0 | | | | | | | | |
| | Colorectal | 29 | 9,576 | 30 | 28 | 51 | 24 | 8,666 | 29 | 22 | 32 | 5 | 910 | 90 | 43 | 279 | 59 | 9 | 110 | 0.010 |
| | Esophageal | 5 | 935 | 31 | 24 | 68 | 4 | 811 | 31 | 22 | 78 | 1 | 124 | 33 | | | 3 | −46 | 11 | 0.724 |
| | Gastric | 4 | 475 | 30 | 25 | 45 | 2 | 329 | 28 | 25 | 30 | 2 | 83 | 38 | 30 | 46 | 10 | 0 | 21 | 0.414 |
| | Liver | 1 | 130 | 9 | | | 1 | 130 | 9 | | | 0 | | | | | | | | |
| | Pancreatic | 5 | 493 | 14 | 11 | 20 | 5 | 493 | 14 | 11 | 20 | 0 | | | | | | | | |
| | Rectal | 5 | 979 | 31 | 20 | 126 | 4 | 945 | 26 | 20 | 118 | 1 | 34 | 128 | | | 101 | 7 | 108 | 0.289 |
| | Stomach | 1 | 187 | 9 | | | 1 | 187 | 9 | | | 0 | | | | | | | | |
| **Genitourinary** | All genitourinary | 32 | 6,784 | 24 | 12 | 30 | 30 | 6,496 | 18 | 10 | 30 | 2 | 288 | 97 | 78 | 116 | 72 | −36 | 112 | 0.129 |
| | Bladder | 12 | 2,135 | 14 | 4 | 96 | 12 | 2,135 | 14 | 4 | 96 | 0 | | | | | | | | |
| | Penile | 3 | 517 | 92 | 75 | 116 | 2 | 263 | 84 | 75 | 92 | 1 | 254 | 116 | | | 33 | 24 | 41 | 0.540 |
| | Prostate | 8 | 3,035 | 27 | 15 | 91 | 8 | 3,035 | 27 | 15 | 91 | 0 | | | | | | | | |
| | Renal | 5 | 567 | 10 | 3 | 66 | 4 | 533 | 7 | 3 | 29 | 1 | 34 | 78 | | | 71 | 48 | 75 | 0.289 |
| | Testicular | 3 | 272 | 29 | 12 | 30 | 3 | 272 | 29 | 12 | 30 | 0 | | | | | | | | |
| | Upper urinary | 0 | | | | | 0 | | | | | 0 | | | | | | | | |
| **Gynecological** | All gynecological | 32 | 4,218 | 27 | 16 | 51 | 22 | 2,206 | 21 | 15 | 30 | 10 | 2,012 | 79 | 40 | 129 | 51 | 22 | 90 | 0.017 |
| | Cervical | 14 | 2,145 | 41 | 25 | 88 | 6 | 502 | 29 | 16 | 43 | 8 | 1,913 | 79 | 14 | 119 | 47 | −17 | 94 | 0.137 |
| | Endometrial | 5 | 664 | 16 | 11 | 56 | 4 | 602 | 15 | 10 | 42 | 1 | 62 | 60 | | | 46 | 17 | 50 | 0.289 |
| | Ovarian | 10 | 1,041 | 18 | 13 | 47 | 9 | 1,004 | 15 | 12 | 28 | 1 | 37 | 168 | | | 153 | 78 | 159 | 0.163 |
| | Uterine | 1 | 35 | 21 | | | 1 | 35 | 21 | | | 0 | | | | | | | | |
| | Vulvar | 2 | 63 | 66 | 17 | 114 | 2 | 63 | 66 | 17 | 114 | 0 | | | | | | | | |
| **Head and neck** | All head and neck | 51 | 5,890 | 45 | 32 | 63 | 22 | 3,361 | 31 | 30 | 58 | 29 | 2,529 | 60 | 45 | 90 | 23 | 1 | 55 | 0.025 |
| | Hypopharyngeal | 1 | 97 | 90 | | | 0 | | | | | 1 | 97 | 90 | | | | | | |
| | Laryngeal | 7 | 540 | 60 | 32 | 96 | 3 | 344 | 89 | 35 | 119 | 4 | 196 | 53 | 24 | 89 | −29 | −96 | 56 | 0.596 |
| | Nasopharyngeal | 3 | 413 | 42 | 30 | 910 | 1 | 101 | 42 | | | 2 | 312 | 60 | 30 | 90 | 18 | −12 | 48 | 1.000 |
| | Oral | 18 | 1,995 | 47 | 31 | 73 | 6 | 690 | 31 | 22 | 87 | 12 | 1,305 | 55 | 35 | 90 | 16 | −21 | 59 | 0.541 |
| | Oropharyngeal | 6 | 1,122 | 30 | 25 | 77 | 4 | 1,085 | 30 | 21 | 63 | 2 | 37 | 60 | 30 | 90 | 18 | −34 | 69 | 0.461 |
| | Thyroid | 2 | 179 | 40 | 21 | 60 | 2 | 179 | 40 | 21 | 60 | 0 | | | | | | | | |
| **Hematological** | All hematological | 38 | 5,346 | 21 | 16 | 29 | 33 | 5,102 | 19 | 16 | 26 | 5 | 244 | 42 | 23 | 156 | 24 | 6 | 57 | 0.017 |
| | Leukemia | 10 | 1,028 | 19 | 11 | 34 | 9 | 973 | 16 | 9 | 32 | 1 | 55 | 186 | | | 170 | 151 | 182 | 0.163 |
| | Lymphoma | 17 | 2,517 | 21 | 17 | 28 | 14 | 2,357 | 20 | 16 | 27 | 3 | 160 | 28 | 21 | 42 | 10 | −2 | 25 | 0.184 |
| | Myeloma | 5 | 1,287 | 24 | 6 | 48 | 5 | 1,287 | 24 | 6 | 48 | 0 | | | | | | | | |

*(Continued)*

Table 1. (Continued)

| Body location/system | Cancer site | PI All studies | | | | | PI Studies from high-income countries | | | | | PI Studies from lower-income countries | | | | | PI Difference | | | |
|---|---|---|---|---|---|---|---|---|---|---|---|---|---|---|---|---|---|---|---|---|
| | | k | N | med | LLCI | ULCI | k | N | med | LLCI | ULCI | k | N | med | LLCI | ULCI | Diff | LLCI | ULCI | p |
| **Musculoskeletal** | Sarcoma | 13 | 4,148 | 75 | 29 | 130 | 12 | 4,125 | 60 | 29 | 154 | 1 | 23 | 86 | | | 15 | −158 | 73 | 0.689 |
| **Neurologic/brain** | Neurologic/brain | 5 | 320 | 48 | 9 | 616 | 2 | 192 | 11 | 7 | 15 | 3 | 128 | 270 | 48 | 730 | 258 | 33 | 723 | 0.149 |
| **Respiratory/thoracic** | Lung | 37 | 17,964 | 28 | 22 | 32 | 23 | 5,663 | 22 | 20 | 30 | 14 | 12,301 | 33 | 28 | 59 | 13 | 3 | 31 | 0.023 |
| **Skin** | Melanoma | 9 | 1,353 | 85 | 39 | 334 | 7 | 1,111 | 70 | 25 | 764 | 2 | 242 | 118 | 85 | 150 | 35 | −1,775 | 130 | 0.661 |
| | Nonmelanoma | 5 | 1,511 | 60 | 57 | 476 | 5 | 1,511 | 60 | 57 | 477 | 0 | | | | | | | | |
| **Unknown** | Unknown primary | 1 | 110 | 7 | | | 1 | 110 | 7 | | | 0 | | | | | | | | |

k = number of studies/estimates joined in meta-analysis; N = number of patients; med = pooled median (in days); LLCI and ULCI = lower-level and upper-level confidence interval, respectively;

Diff = difference between lower- and high-income countries based on Wilcoxon rank sum test; PA-associated = pregnancy-associated.

When k = 1, the median from the single located study is reported.

*Breast cancer only includes female breast cancer.

**Upper-gastrointestinal cancers include esophageal cancer, stomach cancer, small bowel cancer, pancreatic cancer, liver cancer, and cancers of the biliary system.

Table 2. Meta-analysis results from studies reporting on the diagnostic interval (DI).

| Body location/system | Cancer site | DI All studies | | | | | DI Studies from high-income countries | | | | | DI Studies from lower-income countries | | | | | DI Difference | | | |
|---|---|---|---|---|---|---|---|---|---|---|---|---|---|---|---|---|---|---|---|---|
| | | k | N | med | LLCI | ULCI | k | N | med | LLCI | ULCI | k | N | med | LLCI | ULCI | Diff | LLCI | ULCI | p |
| **Breast** | All breast* | 56 | 95,053 | 25 | 21 | 28 | 30 | 87,053 | 19 | 11 | 22 | 26 | 8,000 | 55 | 26 | 93 | 32 | 14 | 64 | <0.001 |
| | Breast not PA-associated | 51 | 94,808 | 25 | 21 | 28 | 26 | 86,820 | 21 | 14 | 25 | 25 | 7,988 | 53 | 26 | 87 | 26 | 9 | 57 | <0.001 |
| | Breast PA-associated | 5 | 245 | 8 | 1 | 161 | 4 | 233 | 5 | 1 | 9 | 1 | 12 | 212 | | | 207 | 203 | 211 | 0.289 |
| **Digestive/gastrointestinal** | All digestive/gastrointestinal | 68 | 72,407 | 57 | 45 | 67 | 63 | 71,772 | 59 | 45 | 70 | 5 | 635 | 46 | 30 | 80 | -12 | -36 | 11 | 0.365 |
| | All upper gastrointestinal** | 21 | 6,627 | 43 | 30 | 62 | 21 | 6,627 | 43 | 30 | 62 | 0 | | | | | | | | |
| | Anal | 0 | | | | | 0 | | | | | 0 | | | | | | | | |
| | Colon | 4 | 8,323 | 55 | 36 | 64 | 4 | 8,323 | 55 | 36 | 64 | 0 | | | | | | | | |
| | Colorectal | 42 | 63,862 | 63 | 48 | 78 | 40 | 63,490 | 63 | 48 | 78 | 2 | 372 | 61 | 30 | 91 | -7 | -67 | 53 | 0.658 |
| | Esophageal | 6 | 3,100 | 39 | 24 | 76 | 6 | 3,100 | 39 | 25 | 78 | 0 | | | | | | | | |
| | Gastric | 5 | 1,955 | 46 | 35 | 97 | 3 | 1,712 | 84 | 44 | 101 | 2 | 243 | 39 | 32 | 46 | -43 | -69 | 2 | 0.386 |
| | Liver | 2 | 248 | 19 | 6 | 31 | 2 | 248 | 19 | 6 | 31 | 0 | | | | | | | | |
| | Pancreatic | 10 | 2,968 | 53 | 30 | 70 | 10 | 2,968 | 53 | 30 | 70 | 0 | | | | | | | | |
| | Rectal | 4 | 3,846 | 41 | 36 | 61 | 4 | 3,846 | 41 | 36 | 61 | 0 | | | | | | | | |
| | Stomach | 2 | 306 | 42 | 42 | 42 | 2 | 306 | 42 | 42 | 42 | 0 | | | | | | | | |
| **Genitourinary** | All genitourinary | 37 | 44,107 | 58 | 50 | 77 | 37 | 44,107 | 58 | 50 | 77 | 0 | | | | | | | | |
| | Bladder | 15 | 34,854 | 53 | 34 | 76 | 15 | 34,854 | 53 | 34 | 76 | 0 | | | | | | | | |
| | Penile | 0 | | | | | 0 | | | | | 0 | | | | | | | | |
| | Prostate | 11 | 6,378 | 85 | 57 | 112 | 11 | 6,378 | 85 | 57 | 112 | 0 | | | | | | | | |
| | Renal | 6 | 1,311 | 72 | 38 | 102 | 6 | 1,311 | 72 | 38 | 102 | 0 | | | | | | | | |
| | Testicular | 3 | 238 | 41 | 7 | 53 | 3 | 238 | 41 | 7 | 53 | 0 | | | | | | | | |
| | Upper urinary | 2 | 1,326 | 55 | 49 | 60 | 2 | 1,326 | 55 | 49 | 60 | 0 | | | | | | | | |
| **Gynecological** | All gynecological | 21 | 4,086 | 48 | 40 | 76 | 20 | 3,855 | 46 | 39 | 72 | 1 | 231 | 97 | | | 50 | -38 | 70 | 0.215 |
| | Cervical | 5 | 504 | 75 | 49 | 95 | 4 | 273 | 71 | 44 | 90 | 1 | 231 | 97 | | | 26 | 7 | 54 | 0.289 |
| | Endometrial | 4 | 1,374 | 55 | 34 | 84 | 4 | 1,374 | 55 | 34 | 84 | 0 | | | | | | | | |
| | Ovarian | 10 | 2,134 | 45 | 34 | 86 | 10 | 2,134 | 45 | 34 | 86 | 0 | | | | | | | | |
| | Uterine | 0 | | | | | 0 | | | | | 0 | | | | | | | | |
| | Vulvar | 1 | 14 | 43 | | | 1 | 14 | 43 | | | 0 | | | | | | | | |
| **Head and neck** | All head and neck | 29 | 4,364 | 35 | 26 | 50 | 18 | 3,108 | 37 | 23 | 85 | 11 | 1,256 | 30 | 21 | 48 | -7 | -45 | 11 | 0.312 |
| | Hypopharyngeal | 0 | | | | | 0 | | | | | 0 | | | | | | | | |
| | Laryngeal | 2 | 151 | 48 | 32 | 63 | 0 | | | | | 1 | 151 | 48 | | | | | | |
| | Nasopharyngeal | 1 | 307 | 90 | | | 0 | | | | | 1 | 307 | 90 | | | | | | |
| | Oral | 11 | 1,303 | 35 | 21 | 38 | 6 | 766 | 36 | 24 | 39 | 5 | 537 | 30 | 16 | 44 | -6 | -21 | 14 | 0.358 |
| | Oropharyngeal | 2 | 726 | 54 | 25 | 83 | 1 | 703 | 83 | | | 1 | 23 | 25 | | | | | | |
| | Thyroid | 0 | | | | | 0 | | | | | 0 | | | | | | | | |
| **Hematological** | All hematological | 40 | 10,178 | 71 | 52 | 85 | 36 | 9,958 | 71 | 53 | 87 | 4 | 220 | 63 | 14 | 363 | 0 | -57 | 273 | 1.000 |
| | Leukemia | 11 | 1,371 | 30 | 13 | 87 | 10 | 1,316 | 23 | 12 | 69 | 1 | 55 | 372 | | | 351 | 270 | 363 | 0.154 |
| | Lymphoma | 17 | 3,160 | 69 | 44 | 82 | 14 | 2,995 | 70 | 51 | 85 | 3 | 165 | 42 | 14 | 84 | -24 | -67 | 23 | 0.344 |
| | Myeloma | 7 | 5,162 | 83 | 47 | 145 | 7 | 5,162 | 83 | 47 | 145 | 0 | | | | | | | | |

*(Continued)*

**Table 2.** (Continued)

| Body location/system | Cancer site | DI All studies | | | | | DI Studies from high-income countries | | | | | DI Studies from lower-income countries | | | | | DI Difference | | | |
|---|---|---|---|---|---|---|---|---|---|---|---|---|---|---|---|---|---|---|---|---|
| | | k | N | med | LLCI | ULCI | k | N | med | LLCI | ULCI | k | N | med | LLCI | ULCI | Diff | LLCI | ULCI | p |
| **Musculoskeletal** | Sarcoma | 6 | 297 | 109 | 78 | 229 | 6 | 297 | 109 | 79 | 229 | 0 | | | | | | | | |
| **Neurologic/brain** | Neurologic/brain | 3 | 300 | 28 | 6 | 29 | 2 | 237 | 18 | 6 | 29 | 1 | 63 | 28 | | | 11 | −1 | 22 | 1.000 |
| **Respiratory/thoracic** | Lung | 25 | 31,904 | 43 | 34 | 53 | 22 | 19,348 | 47 | 35 | 54 | 3 | 12,556 | 34 | 30 | 40 | −12 | −53 | 5 | 0.209 |
| **Skin** | Melanoma | 11 | 1,938 | 29 | 18 | 35 | 11 | 1,938 | 29 | 18 | 35 | 0 | | | | | | | | |
| | Nonmelanoma | 0 | | | | | 0 | | | | | 0 | | | | | | | | |
| **Unknown** | Unknown primary | 2 | 334 | 25 | 15 | 35 | 2 | 334 | 25 | 15 | 35 | 0 | | | | | | | | |

k = number of studies/estimates joined in meta-analysis; N = number of patients; med = pooled median (in days); LLCI and ULCI = lower-level and upper-level confidence interval, respectively;

Diff = difference between lower- and high-income countries based on Wilcoxon rank sum test; PA-associated = pregnancy-associated.

When k = 1, the median from the single located study is reported.

*Breast cancer only includes female breast cancer.

**Upper-gastrointestinal cancers include esophageal cancer, stomach cancer, small bowel cancer, pancreatic cancer, liver cancer, and cancers of the biliary system.

**Table 3. Meta-analysis results from studies reporting on the treatment interval (TI).**

| Body location/system | Cancer site | TI All studies | | | | | TI Studies from high-income countries | | | | | TI Studies from lower-income countries | | | | | TI Difference | | | |
|---|---|---|---|---|---|---|---|---|---|---|---|---|---|---|---|---|---|---|---|---|
| | | k | N | med | LLCI | ULCI | k | N | med | LLCI | ULCI | k | N | med | LLCI | ULCI | Diff | LLCI | ULCI | p |
| **Breast** | All breast* | 99 | 1,711,613 | 29 | 27 | 31 | 70 | 1,700,440 | 30 | 27 | 32 | 29 | 11,173 | 28 | 22 | 40 | 1 | −5 | 8 | 0.857 |
| | Breast not PA-associated | 95 | 1,711,380 | 30 | 27 | 32 | 66 | 1,700,207 | 31 | 27 | 32 | 29 | 11,173 | 28 | 22 | 40 | 2 | −6 | 8 | 0.994 |
| | Breast PA-associated | 4 | 233 | 20 | 19 | 24 | 4 | 233 | 20 | 19 | 24 | 0 | | | | | | | | |
| **Digestive/ gastrointestinal** | All digestive/ gastrointestinal | 105 | 1,126,671 | 23 | 20 | 31 | 95 | 1,046,841 | 24 | 20 | 32 | 10 | 79,830 | 20 | 13 | 37 | −3 | −15 | 7 | 0.512 |
| | All upper gastrointestinal** | 19 | 94,830 | 40 | 22 | 51 | 18 | 94,706 | 41 | 21 | 54 | 1 | 124 | 28 | | | −13 | −137 | 14 | 0.648 |
| | Anal | 11 | 12,546 | 33 | 32 | 34 | 11 | 12,546 | 33 | 32 | 34 | 0 | | | | | | | | |
| | Colon | 19 | 170,171 | 16 | 7 | 23 | 18 | 131,038 | 15 | 7 | 23 | 1 | 39,133 | 21 | | | 8 | −43 | 17 | 0.645 |
| | Colorectal | 68 | 1,003,355 | 18 | 14 | 22 | 61 | 923,732 | 18 | 14 | 23 | 7 | 79,623 | 19 | 10 | 30 | −2 | −13 | 8 | 0.809 |
| | Esophageal | 6 | 9,870 | 44 | 30 | 63 | 5 | 9,746 | 45 | 35 | 66 | 1 | 124 | 28 | | | −16 | −40 | −5 | 0.242 |
| | Gastric | 7 | 15,940 | 32 | 16 | 89 | 5 | 15,857 | 32 | 19 | 65 | 2 | 83 | 82 | 14 | 150 | 38 | −58 | 133 | 1.000 |
| | Liver | 3 | 15,495 | 50 | 40 | 51 | 3 | 15,495 | 50 | 40 | 51 | 0 | | | | | | | | |
| | Pancreatic | 8 | 69,369 | 20 | 15 | 28 | 8 | 69,369 | 20 | 15 | 28 | 0 | | | | | | | | |
| | Rectal | 9 | 68,805 | 26 | 18 | 49 | 6 | 28,623 | 33 | 19 | 48 | 3 | 40,182 | 22 | 11 | 60 | −5 | −44 | 44 | 0.795 |
| | Stomach | 0 | | | | | 0 | | | | | 0 | | | | | | | | |
| **Genitourinary** | All genitourinary | 49 | 1,370,236 | 57 | 45 | 66 | 48 | 1,370,202 | 59 | 45 | 67 | 1 | 34 | 34 | | | −24 | −146 | 34 | 0.457 |
| | Bladder | 11 | 26,632 | 50 | 5 | 65 | 11 | 26,632 | 50 | 5 | 65 | 0 | | | | | | | | |
| | Penile | 1 | 13,283 | 27 | | | 1 | 13,283 | 27 | | | 0 | | | | | | | | |
| | Prostate | 27 | 1,060,169 | 75 | 61 | 87 | 27 | 1,060,169 | 75 | 61 | 87 | 0 | | | | | | | | |
| | Renal | 9 | 269,965 | 0 | 0 | 22 | 8 | 269,931 | 0 | 0 | 2 | 1 | 34 | 34 | | | 34 | 10 | 34 | 0.250 |
| | Testicular | 0 | | | | | 0 | | | | | 0 | | | | | | | | |
| | Upper urinary | 1 | 187 | 45 | | | 1 | 187 | 45 | | | 0 | | | | | | | | |
| **Gynecological** | All gynecological | 33 | 136,330 | 46 | 38 | 54 | 22 | 130,833 | 42 | 34 | 48 | 11 | 5,497 | 69 | 28 | 107 | 28 | 11 | 58 | 0.015 |
| | Cervical | 13 | 6,394 | 69 | 45 | 108 | 4 | 1,002 | 43 | 11 | 117 | 9 | 5,392 | 71 | 63 | 109 | 29 | −48 | 76 | 0.316 |
| | Endometrial | 5 | 120,408 | 38 | 17 | 47 | 4 | 120,346 | 41 | 16 | 48 | 1 | 62 | 24 | | | −19 | −24 | 9 | 0.724 |
| | Ovarian | 3 | 145 | 13 | 1 | 17 | 2 | 102 | 7 | 1 | 13 | 1 | 43 | 17 | | | 10 | 4 | 16 | 0.540 |
| | Uterine | 11 | 9,369 | 46 | 37 | 53 | 11 | 9,369 | 46 | 37 | 53 | 0 | | | | | | | | |
| | Vulvar | 1 | 14 | 18 | | | 1 | 14 | 18 | | | 0 | | | | | | | | |
| **Head and neck** | All head and neck | 77 | 126,777 | 33 | 32 | 37 | 69 | 125,693 | 33 | 32 | 37 | 8 | 1,084 | 41 | 20 | 65 | 2 | −11 | 25 | 0.764 |
| | Hypopharyngeal | 3 | 4,984 | 32 | 27 | 37 | 3 | 4,984 | 32 | 27 | 37 | 0 | | | | | | | | |
| | Laryngeal | 8 | 35,318 | 26 | 22 | 42 | 6 | 34,720 | 26 | 23 | 38 | 2 | 598 | 37 | 18 | 56 | 5 | −25 | 34 | 1.000 |
| | Nasopharyngeal | 1 | 101 | 39 | | | 1 | 101 | 39 | | | 0 | | | | | | | | |
| | Oral | 19 | 29,047 | 30 | 23 | 53 | 16 | 28,846 | 30 | 23 | 53 | 3 | 201 | 55 | 20 | 103 | 25 | −29 | 76 | 0.342 |
| | Oropharyngeal | 8 | 14,459 | 34 | 29 | 58 | 8 | 14,459 | 34 | 29 | 58 | 0 | | | | | | | | |
| | Thyroid | 4 | 6,256 | 165 | 67 | 502 | 4 | 6,256 | 165 | 67 | 502 | 0 | | | | | | | | |

(*Continued*)

Table 3. (Continued)

| Body location/system | Cancer site | TI All studies | | | | | TI Studies from high-income countries | | | | | TI Studies from lower-income countries | | | | | TI Difference | | | |
|---|---|---|---|---|---|---|---|---|---|---|---|---|---|---|---|---|---|---|---|---|
| | | k | N | med | LLCI | ULCI | k | N | med | LLCI | ULCI | k | N | med | LLCI | ULCI | Diff | LLCI | ULCI | p |
| **Hematological** | All hematological | 18 | 18,764 | 22 | 8 | 32 | 17 | 18,735 | 21 | 8 | 28 | 1 | 29 | 37 | | | 16 | −2,063 | 35 | 0.385 |
| | Leukemia | 3 | 4,179 | 4 | 3 | 8 | 3 | 4,179 | 4 | 3 | 8 | 0 | | | | | | | | |
| | Lymphoma | 11 | 14,452 | 22 | 12 | 29 | 11 | 14,452 | 22 | 12 | 29 | 0 | | | | | | | | |
| | Myeloma | 1 | 53 | 2 | | | 1 | 53 | 2 | | | 0 | | | | | | | | |
| **Musculoskeletal** | Sarcoma | 11 | 13,352 | 21 | 21 | 22 | 10 | 13,329 | 21 | 21 | 22 | 1 | 23 | 23 | | | 2 | −3 | 3 | 0.252 |
| **Neurologic/brain** | Neurologic/brain | 2 | 65 | 26 | 25 | 26 | 0 | | | | | 2 | 65 | 26 | 255 | 26 | | | | |
| **Respiratory/thoracic** | Lung | 60 | 557,374 | 32 | 27 | 35 | 53 | 546,499 | 33 | 30 | 42 | 7 | 10,875 | 20 | 6 | 25 | −21 | −36 | −9 | 0.002 |
| **Skin** | Melanoma | 9 | 105,326 | 31 | 28 | 45 | 7 | 105,255 | 30 | 24 | 39 | 2 | 71 | 63 | 32 | 95 | 31 | −20 | 82 | 0.188 |
| | Non-melanoma | 0 | | | | | 0 | | | | | 0 | | | | | | | | |
| **Unknown** | Unknown primary | 0 | | | | | 0 | | | | | 0 | | | | | | | | |

k = number of studies/estimates joined in meta-analysis; N = number of patients; med = pooled median (in days); LLCI and ULCI = lower-level and upper-level confidence interval, respectively;

Diff = difference between lower- and high-income countries based on Wilcoxon rank sum test; PA-associated = pregnancy-associated.

When k = 1, the median from the single located study is reported.

*Breast cancer only includes female breast cancer.

**Upper-gastrointestinal cancers include esophageal cancer, stomach cancer, small bowel cancer, pancreatic cancer, liver cancer, and cancers of the biliary system.

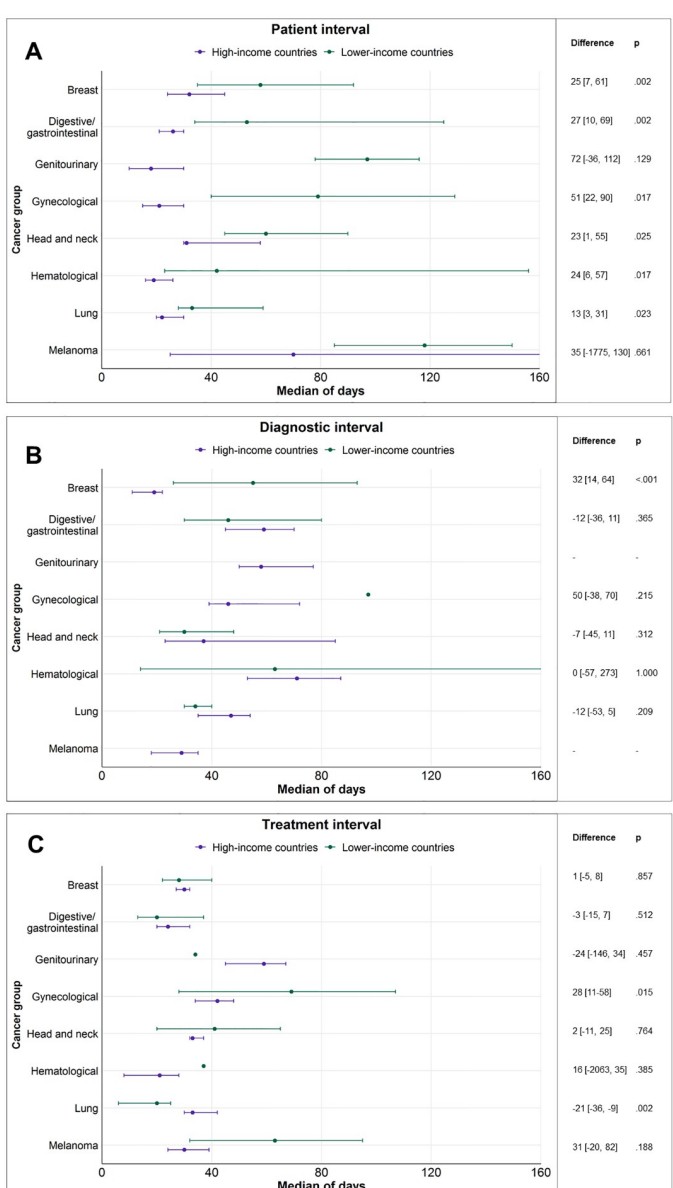

**Fig 3. Pooled median duration and 95% CIs for the patient (A), diagnostic (B), and treatment (C) intervals for the main cancer groups and as a function of GNI (classified into high- and lower-income) of the countries represented.** Note: The lack of CI means that the median represents 1 available study only. When the whiskers for the upper limit of the CI are not seen, the limit is higher than the axis maximum (>160). Difference = estimated difference (and 95% CIs in square brackets) between lower- and high-income countries based on Wilcoxon rank sum test. CI, confidence interval; GNI, gross national income.

gynecological cancers ($R^2$ = 31% based on GNI and 29% based on HDI); and 60 days [45 to 90] versus 31 days [30 to 58] for head and neck cancers ($R^2$ = 8% based on GNI and 17% based on HDI). Differences for the rest of the main cancer groups were less pronounced but still significant: 42 days [23 to 156] versus 19 days [16 to 26] for hematological; 50 days [34 to 90] versus 29 days [19 to 45] for breast; and 33 days [28 to 59] versus 22 days [20 to 30] for lung cancers. Results were similar and differences generally more pronounced using the continuous HDI compared to the binary GNI classification (see S6 Table).

Considering patient interval variation within high-income countries only, patient intervals were longer in studies conducted in countries with lower scores on the Policy and Planning ICP for breast (34 days [34 to 52] versus 7 days [7 to 7] for countries with higher scores) and head and neck cancer (60 days [31 to 64] versus 30 days [21 to 35] for countries with higher scores; see S7 Table).

**Diagnostic intervals.** With the exception of breast (k = 25) and head and neck cancers (k = 11), few studies from lower-income countries reported data on the diagnostic interval (k = 14 for all remaining cancer sites). As a result, comparisons between high- and lower-income countries were statistically meaningful only for a handful of sites. Hence, we discuss the results averaged across all countries (see Table 2).

Median diagnostic intervals generally exceeded 1 month for the majority of cancer sites (i.e., the 25th percentile of the distribution of pooled medians was 36 days). There was large variation across the main cancer groups. The longest diagnostic intervals were observed for hematological (71 days [52 to 85]), genitourinary (58 days [50 to 77]), digestive/gastrointestinal (57 days [45 to 67]), and gynecological cancers (48 days [40 to 76]), followed by lung (43 days [34 to 53]), head and neck cancers (35 days [26 to 50]), skin melanoma (29 days [18 to 35]), and breast cancers (25 days [21 to 28]).

There was significant variation for the specific hematological malignancies with a much shorter pooled diagnostic interval for leukemias (30 days [13 to 87]) compared to lymphomas (69 days [44 to 82]) and myelomas (83 days [47 to 145]). In the case of genitourinary cancers, diagnostic intervals were longest for prostate (85 days [57 to 112]) and renal cancers (72 days [38 to 102]). For digestive/gastrointestinal cancers, pooled diagnostic intervals were longer for colorectal cancers (63 days [48 to 78]) compared to upper gastrointestinal cancers (43 days [30 to 62]). Within gynecological cancers, the longest diagnostic intervals were observed for cervical cancer (75 days [49 to 95]).

Regarding the few documented differences between high- and lower-income countries, the diagnostic interval for breast cancer was significantly longer in lower-income countries (53 days [26 to 87] versus 21 days [14 to 25], with $R^2$ = 25% based on GNI and 29% based on HDI; see Table 2 and S8 Table).

Considering only high-income countries, diagnostic intervals were longer in studies conducted in countries with lower scores on the ICP Policy and Planning for digestive/gastrointestinal cancers (84 days [64 to 92] versus 72 days [67 to 97] for countries with higher scores; see S7 Table).

**Treatment intervals.** With the exception of breast (k = 29), gynecological (k = 11), and digestive/gastrointestinal cancers (k = 10), the number of studies from lower-income countries was again small (k = 22 for all remaining cancer sites), so we discuss the results averaged across all countries (Table 3).

Treatment intervals were relatively more homogeneous than the other intervals and generally varied between 20 and 50 days for the main cancer groups. The longest treatment intervals were observed for genitourinary cancers (57 days [45 to 66]), followed by gynecological cancers (46 days [38 to 54]), head and neck cancers (33 days [32 to 37]), lung (32 days [27 to 35]), melanoma (31 days [28 to 45]), and breast cancers (29 days [27 to 31]). Shorter pooled treatment intervals were observed for digestive/gastrointestinal (23 days [20 to 31]) and hematological (22 days [8 to 32]) malignancies.

Within genitourinary cancers, the longest treatment intervals were observed for prostate cancer (75 days [61 to 87]) and within gynecological cancers for cervical cancer (69 days [45 to 108]). Within head and neck cancers, treatment intervals for thyroid cancers were notably longer than for the other specific cancer sites (165 days [67 to 502], compared to pooled estimates between 26 and 34 days for the rest of the cancer sites). Within digestive/gastrointestinal

cancers, treatment intervals were longer for upper gastrointestinal cancers (40 days [22 to 51]) than for colorectal cancers (18 days [14 to 22]).

The significant differences observed between lower- and high-income countries were that treatment intervals were longer for gynecological (69 days [28 to 107] versus 42 days [34 to 48], $R^2 = 72\%$ for GNI and $R^2 = 64\%$ for HDI) but shorter for lung cancers (20 days [6 to 25] versus 33 days [30 to 42], $R^2 = 20\%$ for GNI and $R^2 = 26\%$ for HDI) in lower-income countries (see Table 3 and S9 Table).

Considering only high-income countries, treatment intervals were longer in studies conducted in countries with higher scores on the ICP Policy and Planning for genitourinary cancers (83 days [65 to 107] versus 57 days [50 to 60] for countries with lower scores; see S7 Table) and specifically for prostate cancer (83 days [77 to 107] versus 60 days [57 to 61] for countries with lower scores).

**Relative contribution of intervals.** For this analysis, we only considered studies that reported on the duration of all 3 intervals in the same patients (see S10 Table). The diagnostic interval was significantly longer than both the patient and treatment intervals for hematological cancers (DI/PI ratio = 3.3 [95% CI 1.2 to 8.8]; DI/TI ratio = 10.3 [3.0 to 35.4]), colorectal cancer (DI/PI ratio = 2.8 [1.0 to 7.6]; DI/TI ratio = 3.4 [2.1 to 5.7]), and lung cancer (DI/PI ratio = 2.1 [1.4 to 3.0]; DI/TI ratio = 1.6 [1.0 to 2.5]). The diagnostic interval was also longer than the treatment interval for gynecological cancers (DI/TI ratio = 3.0 [2.1 to 4.5]).

**Sensitivity analyses.** The intervals excluding high risk of bias studies for 11 cancer groups are reported in S11 Table. In most cases, intervals remained largely unchanged (i.e., change up to ±1 day for 5, 8, and 9 cancer groups on the patient, diagnostic, and treatment intervals, respectively) or changed up to 6 days (for 3, 3, and 0 cancer groups on the patient, diagnostic, and treatment intervals, respectively). Exceptions included patient intervals for the rarer cancer sites that were significantly reduced after excluding studies with high risk of bias. To illustrate, in the case of neurologic/brain cancers, the pooled patient interval was reduced with 33 days from 48 days [9 to 616] to 15 days [7 to 730]; in the case of sarcoma, it was reduced with 30 days from 75 days [29 to 130] to 45 days [25 to 141]; and in the case of melanoma, it was reduced with 15 days from 85 days [39 to 334] to 70 days [25 to 217]. The treatment interval for hematological malignancies was also reduced with 10 days from 22 days [8 to 32] to 12 days [6 to 25].

Results based only on studies that reported medians (excluding studies that reported the means only) are displayed in S12 Table for 11 cancer groups. The majority of intervals remained largely unchanged (i.e., change up to ±1 day for 6, 8, and 8 cancer groups on the patient, diagnostic, and treatment intervals, respectively) or changed up to 7 days (for 2, 3, and 3 cancer groups on the patient, diagnostic, and treatment intervals, respectively). Exceptions included the patient intervals for sarcoma, melanoma, and genitourinary cancers, which were reduced after the exclusion of studies reporting means only, with 15 days in the case of sarcoma (from 75 days [29 to 130] to 60 days [27 to 114]), 15 days in the case of melanoma (from 85 days [39 to 334] to 70 days [25 to 170]), and 10 days in the case of genitourinary cancers (from 24 days [12 to 30] to 14 days [8 to 30]).

## Discussion

To our knowledge, this is the first review to offer meta-analytical estimates of the pooled median duration of the patient, diagnostic, and treatment intervals in adult patients with diverse types of cancer. The results of this descriptive and comparative study can be useful in the monitoring and evaluation of early diagnosis efforts and the design of interventions to strengthen early diagnosis and timely treatment [2]. The broad scope of the review also

provides useful information regarding the amount of evidence available for the different cancer sites and can help set research priorities in the field [18]. Whereas all 3 intervals were frequently reported for patients with breast, lung, colorectal, and head and neck cancer, fewer studies were available for the other cancer sites. Importantly, only 28% of the identified articles reported data from lower-income countries and mostly on the patient interval.

The review revealed some striking differences between high- and lower-income countries in the duration of the patient interval. Pooled patient intervals were relatively more homogeneous across most cancer sites in studies from high-income countries, showing that at least half of patients with symptomatic cancer present to a healthcare professional within a month of symptom onset (e.g., pooled medians generally between 15 and 31 days). Results revealed that patient intervals in lower-income countries were consistently 1.5 to 4 times longer, ranging generally between 1 and 3 months. These results are in accordance with those from previous reviews focused on lower-income countries [20,23], which were mostly based on studies on breast and childhood cancer. The literature on barriers to help-seeking indicates that low cancer symptom recognition and negative beliefs about cancer are likely universal predictors of longer patient intervals [32,37]. However, there are unique factors in lower-income contexts such as low health literacy, the use of alternative medicine, female-specific barriers (e.g., the need for family permission to seek help), strong negative stigma of cancer treatment, and financial and access barriers that may delay help-seeking [37]. A recent review of 25 interventions conducted in lower- and middle-income countries found that some were effective at increasing knowledge (e.g., about cancer in general, early detection, or signs and symptoms) but concluded that interventions are needed focusing on more clinically relevant outcomes [38].

Fewer studies were available from lower-income countries, especially reporting on the diagnostic and treatment intervals. Information on these intervals came mostly from high-income countries that have powerful health information systems in place to record, monitor, and analyze such information (e.g., population-based cancer registries, national healthcare databases, and complete and often easily accessible medical records). For example, studies reporting treatment intervals were mostly based on such information systems and frequently had very large sample sizes (in the thousands), offering representative data. The expansion and creation of cancer registries or large cancer epidemiological databases in lower-income countries as an investment in national cancer control planning is one of the priorities suggested to reduce cancer care disparities worldwide [22]. There were fewer differences between high- and lower-income countries on the diagnostic and treatment intervals at least partially due to lack of enough data from lower-income countries for comparison. Nevertheless, we documented significantly longer diagnostic intervals for breast cancer and longer treatment intervals for gynecological cancers in lower-income countries, both of which could be contributing to the lower survival of these cancers (especially cervical cancer) in lower-income countries [39].

Overall, the longest diagnostic intervals were observed for hematological, genitourinary, digestive/gastrointestinal, and gynecological malignancies. The reasons for such long times from the first consultation to diagnosis are likely multiple. The cancers with longest diagnostic intervals included several cancers classified as "difficult to suspect" (e.g., myeloma, pancreatic) and "intermediate" in difficulty to suspect (e.g., colorectal, lymphoma) [40]. Cancers that are difficult to suspect are characterized by presentation with nonspecific symptoms and the frequent need for multiple consultations before cancer is suspected and diagnosed (in >30% of patients). For cancers that are considered intermediate, some patients present with specific "alarm" symptoms but other may present atypically (between 10% and 30% of patients have multiple consultations before diagnosis). Gynecological cancers, especially endometrial and ovarian cancer, are also frequently characterized by nonspecific symptoms that can be due to benign causes, rendering early diagnosis and treatment difficult [41].

The longest treatment intervals were observed for genitourinary cancers, driven in particular by prostate cancer. Prostate cancer is a relatively slow-growing malignancy and watchful waiting is a standard strategy in low-risk prostate cancer to decrease risk of overtreatment. In addition, evidence suggests that treatment delays up to 3 months can be considered safe for all localized prostate cancer patients [42,43]. Thus, one hypothesis that could be tested in future research is that the long treatment intervals for prostate cancer are due to many patients undergoing "watchful waiting," although concerns about treatment morbidity or stigma could also play a role in some contexts. In addition, studies from high-income countries with lower scores on the ICP on Policy and Planning (e.g., Italy, USA, Spain) reported shorter treatment intervals for prostate cancer than studies from countries with higher scores (e.g., Australia, the Netherlands, Canada, Germany). This could also be due to higher implementation of watchful waiting for prostate cancer or controlled treatment delays in certain contexts.

Unexpectedly, lung cancer treatment intervals were found to be lower in lower-income countries. This could be due to the higher access to last-generation biological and precision therapies in higher-income contexts [44]. Such therapies require genetic testing for treatment selection, which could increase the time elapsed between diagnosis and treatment in high-income countries where such therapies may be more likely to be available.

The additional analysis focused on high-income countries included in the ICP [33] revealed that the existence and implementation of diverse cancer-directed policies is related to shorter patient and diagnostic intervals for some cancers. In the case of the patient interval for breast and head and neck cancers and the diagnostic interval for digestive/gastrointestinal cancers, lower Policy and Planning scores on the ICP were associated with longer intervals. These results suggest that in high-income contexts, the implementation of cancer-directed policies such as national cancer plans including strategies for primary prevention and early detection of cancer [45] could have positive effects on diagnostic delays. Whereas it is not clear what policies exactly may be driving these effects and having in mind that these results are at best preliminary, they offer much needed evidence regarding the potential effects of cancer policies on relevant outcomes [45].

Whereas the grouping of specific cancer sites into general main groups has been useful for descriptive and comparative purposes, it is also limited. The specific cancer sites may present unique challenges and circumstances when it comes to diagnosis and treatment, something that is also reflected in the variation of the pooled intervals within the main cancer groups considered. To take hematological malignancies as an example, the pooled median diagnostic interval for this group was 71 days [52 to 85]. Disaggregating the data further showed very different diagnostic intervals for leukemias (30 days [13 to 87]) compared to lymphomas (69 days [44 to 82]) and myelomas (83 days [47 to 145]). However, even within these more specific groups, there could be large variation in the clinical manifestation and diagnostic process depending on the type of cancer. To illustrate, in a study based on the UK's Hematological Malignancy Research Network, the median duration of the diagnostic interval was 13 days for acute lymphocytic and 10 days for acute myeloid leukemia but 42 days for chronic lymphocytic and 9 days for chronic myeloid leukemia [46]. Discussing the unique diagnostic and treatment circumstances of all cancer sites reported is beyond the scope and possibilities of the review; it is, however, something readers should bear in mind when interpreting our results.

Strengths of the review include the large number of studies identified without country or region restrictions and the use of a validated methodology for the meta-analytic combination of medians. Limitations of the review include the inherent complexity and many possible biases in the measurement of time points and intervals in the cancer treatment pathway [4]. The extent of these is at least partially reflected in the Aarhus checklist scores assigned to each article. Our sensitivity analysis showed that the removal of studies with highest potential for

bias did not substantially affect pooled estimates for the more common cancers. It did, however, change the estimates for some rarer cancers for which there were fewer studies available (e.g., neurologic/brain cancers, sarcoma, and melanoma), and we think that these study quality-adjusted estimates should be considered more reliable. Overall, the low proportion of studies that received a low-risk score on the Aarhus checklist confirms that further efforts are needed to standardize the measurement and reporting of delay intervals [20].

An additional limitation to consider is survivor bias, which is especially relevant for studies using patient interviews and questionnaires. This method of data collection was also especially frequent in studies conducted in lower-income countries that reported on patient intervals. Survivor bias is a type of patient selection bias, where patients dying soon after symptom onset or patients who are too ill to take part in a research study are excluded [47,48]. This could result in biased patient-reported estimates of interval duration and limit generalizability, because patients who die shortly after diagnosis or are too ill to participate may have atypical interval duration. Studies using medical records are less prone to selection and recall biases [47]; however, they have other limitations (e.g., it is assumed that the information recorded during the consultation is complete and accurate, which may not be the case) [48].

Another limitation is that we did not consider the subintervals that compose the diagnostic interval (e.g., primary care interval, referral to diagnosis interval) [4]. We wanted to make comparisons across countries with very different health systems, and we preferred to focus on more generalizable measures of intervals. Future reviews should consider the subintervals that compose the patient (e.g., appraisal versus help-seeking interval) and diagnostic (e.g., primary care interval versus referral to diagnosis interval) intervals to offer a more comprehensive understanding of the patient journey to diagnosis in different contexts. Finally, we did not differentiate between middle-, lower-middle, and low-income countries and grouped them together as "lower-income" economies due to the relatively small number of studies available. However, previous reviews show that there may be important differences in interval duration within this group [20].

In an effort to reduce publication bias, we searched several databases that contain grey literature and considered publications in multiple languages. However, because the data pooled into meta-analysis is descriptive and not based on significance testing, formal tests for publication bias (e.g., funnel plots) could not be performed.

Cancer is a leading cause of death worldwide, and reducing diagnostic and treatment delays could help improve survival and other patient outcomes. This systematic review identified the types of cancer and contexts where diagnosis and treatment initiation may take the longest. These results can be useful to set research priorities and identify areas most in need of interventions to strengthen early diagnosis and timely treatment. Our results also highlight the global disparities in timely diagnosis and treatment. Efforts should be made to reduce help-seeking times for cancer symptoms in lower-income countries.

## Supporting information

**S1 Table. List of excluded articles with reasons for exclusion.**
(XLSX)

**S2 Table. List of included articles.**
(XLSX)

**S3 Table. Number of publications (N) in which each respective country is represented.**
Note: The Total does not equal the number of included publications in the review (410) because 11 publications reported data from multiple countries.
(XLSX)

**S4 Table. Detailed information about included articles.**
(XLSX)

**S5 Table. Aarhus checklist ratings.**
(XLSX)

**S6 Table. Pooled medians for the patient interval as a function of country socioeconomic indicators.** Note: k = number of studies/estimates; N = number of patients; med = pooled median; LLCI and ULCI = lower level and upper level confidence interval, respectively; Difference = pooled median for high-income countries minus pooled median for lower-income countries; Breast cancer excludes pregnancy-associated breast cancer; $p$ = $p$-value from a moderator test in a random-effects model; $R^2$ = percentage of variance in the outcome explained by the moderator. When k = 1, the median from the single located study is reported.
(XLSX)

**S7 Table. Pooled medians for the patient, diagnostic, and treatment intervals from studies conducted in high-income countries as a function of the Index of Cancer Preparedness on Policy and Planning (ICP PP).** Note: k = number of studies/estimates; N = number of patients; med = pooled median; LLCI and ULCI = lower level and upper level confidence interval, respectively; Breast cancer excludes pregnancy-associated breast cancer; $R^2$ = percentage of variance in the outcome explained by the moderator in a moderator test from a random-effects model. When k = 1, the median from the single located study is reported. The high vs. lower IPC PP groups were created based on k-means clustering ("natural" grouping of the data).
(XLSX)

**S8 Table. Pooled medians for the diagnostic interval as a function of country socioeconomic indicators.** Note: k = number of studies/estimates; N = number of patients; med = pooled median; LLCI and ULCI = lower level and upper level confidence interval, respectively; Difference = pooled median for high-income countries minus pooled median for lower-income countries; Breast cancer excludes pregnancy-associated breast cancer; $p$ = $p$-value from a moderator test in a random-effects model, $^* < 0.05$, $^{**} < 0.01$, $^{***} < 0.001$; $R^2$ = percentage of variance in the outcome explained by the moderator. When k = 1, the median from the single located study is reported.
(XLSX)

**S9 Table. Pooled medians for the treatment interval as a function of country socioeconomic indicators.** Note: k = number of studies/estimates; N = number of patients; med = pooled median; LLCI and ULCI = lower level and upper level confidence interval, respectively; Difference = pooled median for high-income countries minus pooled median for lower-income countries; Breast cancer excludes pregnancy-associated breast cancer; $p$ = $p$-value from a moderator test in a random-effects model, $^* < 0.05$, $^{**} < 0.01$, $^{***} < 0.001$; $R^2$ = percentage of variance in the outcome explained by the moderator. When k = 1, the median from the single located study is reported.
(XLSX)

**S10 Table. Pooled medians for the patient, diagnostic, and treatment intervals based on studies that reported on all three intervals for the same patients.** Note: k = number of studies/estimates joined in meta-analysis; N = number of patients; med = pooled median; LLCI and ULCI = lower level and upper level confidence interval, respectively; PI/DI = ratio between the median patient and diagnostic interval. DI/TI = ratio between the median diagnostic and treatment interval. N is sometimes different because of missing data for some

patients. All available studies on digestive/genitourinary cancer are colorectal cancer studies.
(XLSX)

**S11 Table. Pooled median duration of the patient, diagnostic, and treatment interval excluding studies with high risk of bias according to the Aarhus checklist score.** Note: k = number of studies/estimates; N = number of patients; med = pooled median; LLCI and ULCI = lower level and upper level confidence interval, respectively; Difference = the pooled median when studies with high risk are excluded minus the pooled median when all studies are included.
(XLSX)

**S12 Table. Pooled median duration of the patient, diagnostic, and treatment interval excluding studies that only reported means.** Note: k = number of studies/estimates; N = number of patients; med = pooled median; LLCI and ULCI = lower level and upper level confidence interval, respectively; Difference = the pooled median when studies reporting the median minus the pooled median when studies reporting the median or mean are included.
(XLSX)

**S1 Text. Search strategy.**
(DOCX)

**S2 Text. Country socioeconomic indicators.**
(DOCX)

**S3 Text. Aarhus statement checklist—Short.**
(DOCX)

**S4 Text. PRISMA checklist.**
(DOCX)

## Acknowledgments

We thank Dr. Yasmina Okan for her participation in the formulation of the review protocol and the shortened form of the Aarhus checklist. We thank Dr. Elena Salamanca-Fernández for her help with abstract screening.

Where authors are identified as personnel of the International Agency for Research on Cancer/World Health Organization, the authors alone are responsible for the views expressed in this article and they do not necessarily represent the decisions, policy, or views of the International Agency for Research on Cancer/World Health Organization.

## Author Contributions

**Conceptualization:** Dafina Petrova, Ana Ching-López, Maria José Sánchez.

**Data curation:** Dafina Petrova, Zuzana Špacírová, Nicolás Francisco Fernández-Martínez, Ana Ching-López, Dunia Garrido, Daniel Redondo-Sánchez, Camila Higueras-Callejón.

**Formal analysis:** Dafina Petrova, Zuzana Špacírová, Nicolás Francisco Fernández-Martínez, Ana Ching-López, Dunia Garrido, Miguel Rodríguez-Barranco.

**Funding acquisition:** Maria José Sánchez.

**Investigation:** Dafina Petrova, Zuzana Špacírová, Nicolás Francisco Fernández-Martínez, Ana Ching-López, Dunia Garrido, Miguel Rodríguez-Barranco, Marina Pollán, Daniel Redondo-Sánchez, Carolina Espina, Camila Higueras-Callejón.

**Methodology:** Dafina Petrova, Zuzana Špacírová, Nicolás Francisco Fernández-Martínez, Ana Ching-López, Dunia Garrido, Miguel Rodríguez-Barranco, Marina Pollán, Daniel Redondo-Sánchez, Carolina Espina, Maria José Sánchez.

**Project administration:** Dafina Petrova, Maria José Sánchez.

**Resources:** Dafina Petrova, Daniel Redondo-Sánchez, Camila Higueras-Callejón.

**Software:** Dafina Petrova, Daniel Redondo-Sánchez.

**Supervision:** Miguel Rodríguez-Barranco, Marina Pollán.

**Validation:** Dafina Petrova.

**Visualization:** Daniel Redondo-Sánchez.

**Writing – original draft:** Dafina Petrova.

**Writing – review & editing:** Dafina Petrova, Zuzana Špacírová, Nicolás Francisco Fernández-Martínez, Ana Ching-López, Dunia Garrido, Miguel Rodríguez-Barranco, Marina Pollán, Daniel Redondo-Sánchez, Carolina Espina, Camila Higueras-Callejón, Maria José Sánchez.

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
