## [Editor Report · Decision Letter 0]

25 Mar 2022

Dear Dr Petrova, 

Thank you for submitting your manuscript entitled "The patient, diagnostic, and treatment intervals in adult cancer patients: a systematic review and meta-analysis with country socio-economic indicators" for consideration by PLOS Medicine.

Your manuscript has now been evaluated by the PLOS Medicine editorial staff and I am writing to let you know that we would like to send your submission out for external peer review.

Please re-submit your manuscript within two working days, i.e. by Mar 29 2022 11:59PM.

Kind regards,

Beryne Odeny

PLOS Medicine

---

## [Decision Letter · Decision Letter 1]

12 May 2022

Dear Dr. Petrova,

Thank you very much for submitting your manuscript "The patient, diagnostic, and treatment intervals in adult cancer patients: a systematic review and meta-analysis with country socio-economic indicators" (PMEDICINE-D-22-00994R1) for consideration at PLOS Medicine. 

[LINK]

In light of these reviews, I am afraid that we will not be able to accept the manuscript for publication in the journal in its current form, but we would like to consider a revised version that addresses the reviewers' and editors' comments. Obviously we cannot make any decision about publication until we have seen the revised manuscript and your response, and we plan to seek re-review by one or more of the reviewers. 

We expect to receive your revised manuscript by Jun 02 2022 11:59PM. Please email us (plosmedicine@plos.org) if you have any questions or concerns.

We look forward to receiving your revised manuscript. 

Sincerely,

Beryne Odeny, 

PLOS Medicine

plosmedicine.org

1) Please revise your title according to PLOS Medicine's style. Your title must be nondeclarative and not a question. It should begin with main concept if possible. Please place the study design (for example, “A modelling study”) in the subtitle (i.e., after a colon). Please consider “Patient presentation, diagnostic, and treatment intervals in adult patients with cancer across various country-level socioeconomic indicators ….: A systematic review and meta-analysis”

2) Abstract:

a) Please report your abstract according to PRISMA for abstracts, following the PLOS Medicine abstract structure (Background, Methods and Findings, Conclusions) http://www.plosmedicine.org/article/info:doi/10.1371/journal.pmed.1001419 .

i) Please rename the “Introduction” to “Background.”

ii) Please rename the “Methods and Results” to “Methods and findings.” 

b) Please include the eligibility criteria

c) Please outline all the databases you searched

d) Please ensure that all numbers presented in the abstract are present and identical to numbers presented in the main manuscript text.

e) Please provide both 95% CIs and p values where appropriate.

f) In the last sentence of the Abstract Methods and Findings section, please describe the main limitation(s) of the study's methodology.

4) In line with PLOS Medicine’s guidelines, please update your search to the present time and provide the beginning and end dates of your search.

5) Please avoid use of the term “adult cancer patients” and instead refer to “adult patients with cancer.”

6) Please evaluate evidence of publication bias.

7) Please indicate in the figure captions the meaning of the bars and whiskers in figure 3

8) Figure 2: Please confirm that the appropriate usage rights apply to the use of this map. Please see our guidelines for map images: https://journals.plos.org/plosmedicine/s/figures#loc-maps

9) The terms gender and sex are not interchangeable (as discussed in http://www.who.int/gender/whatisgender/en/ ); please use the appropriate term.

10) References: 

a) Please ensure there is no space between in-text reference call outs. For example, “…community [8,9].”

b) Please ensure that journal name abbreviations consistently match those found in the National Center for Biotechnology Information (NCBI) databases. https://journals.plos.org/plosmedicine/s/submission-guidelines#loc-references. 

c) Please include access dates for all weblinks and ensure that all weblinks are current and accessible, e.g. #2, 25 etc

11) Please remove the “Declaration of interest”, ‘Funding’, and “Data availability statement,” from the end of the main text. In the event of publication, this information will be published as metadata based on your responses to the submission form

Comments from the reviewers:

Reviewer #1: Dear Authors,

Thank you for the opportunity to review this systematic review and meta-analysis examining patient, diagnostic and treatment intervals in adult cancer patients using socioeconomic indicators. This is an interesting study, and you should be commended for your transparency in the methodology and explicitly descriptive intention. I do have several comments for your consideration. The introduction adequately describes the interval framework used throughout the manuscript; however, there is not sufficient background information indicating the contributors to the length of time of these intervals and how these vary across regions of different SES. The mention of these prolonged intervals in the introduction within the context of the COVID-19 pandemic is misleading as it is not pertinent to the scope of the study. If the intention is to describe the intervals as a baseline pre-COVID, this should be mentioned explicitly. 

Regarding the methodology, the systematic literature search should be updated as it is almost two years old, and recent publication pertaining to diagnostic and treatment intervals should be captured. My last overall note relates to the reporting of results. For clarity, when median/pooled median intervals are presented, the lower and upper confidence bounds (i.e., range in days) also need to be reported. 

I must also apologize, as I was not able to see line numbers with the provided text and was therefore not able to provide specificity in some of my remarks. I tried my best to provide direction and copy specific sentences to the body of my response, making it easier to control+F. 

I'd like to thank the authors for taking on this important work and for producing such a well written manuscript. With a few changes and spelling corrections, this review will be a meaningful contribution to the PLOS Medicine collection. Please find my notes below. 

Abstract: 

- Please communicate the information sources for the systematic search by explicitly naming the databases and mentioning that grey literature was also reviewed.

- To improve clarity, consider changing "first presentation" (when describing patient intervals) to "first presentation to health care facility" to align with the definition provided in the body of the text. 

- Identify the number of articles captured from each of the SES regions (ex. of the 329, were 100 HIC and 229 LMIC) and report this in the results. 

- Lower and upper bounds need to be provided for each of the pooled estimates (ex. 20 days [range 1-34])

- Further description of the meta-analytic methods is needed in the abstract. Need to identify the measure that is being used to pool—i.e., describe that pooled median intervals were calculated in units of time (days) with their ranges (lower and upper bound).

- The last sentence of the "Results" in the abstract needs to be reworked as it is unclear if there were few studies available from low-income countries in general, or if this pertains to prostate cancers only. Identifying the distribution of studies captured based on HIC and LMIC (as previously mentioned) will aid clarity. 

Introduction:

- The first sentence of the second paragraph requires citations. 

o ["It is generally expected that longer interval duration is associated with worse cancer outcomes such as stage migration (i.e., later stage at diagnosis), the need for more extensive therapy with higher toxicity, higher recurrence rates, and higher mortality] 

- Additional citations, beyond the systematic review already cited, are required for the following sentence. 

o [Consistent with this, there is evidence to suggest that shorter times to diagnosis are associated with better outcomes in terms of stage at diagnosis and survival for breast, colorectal, head and neck, testicular cancers, and skin melanoma, with less evidence for pancreatic, prostate, and bladder cancers]

- The final sentence of the same paragraph needs regional context to tie in the overarching theme of SES. Are the references representative of both high- and low-income countries? And are the transportable to other populations (i.e., does a four-week delay always have a negative impact in all contexts globally?)

- It is not clear to me why COVID-19 is mentioned in the introduction. The research question is important beyond the context of COVID and there is no link between treatment intervals and COVID throughout the manuscript. Consider adding additional rationale (is this to establish a baseline of delays to make meaningful comparisons as the pandemic plays out?) or moving this section to the discussion. 

- A greater emphasis on what this study contributes to the literature is needed. Brand et al. (ref 13) published a SR on a nearly identical topic, with a similar number of articles captured (n=319) in 2019. 

Methods:

- The search is outdated and needs to be updated. If the authors' goal is to avoid capturing articles reporting on treatment intervals during COVID-19 times, consider adding COVID-19 as a "NOT" term in the search strategy. Restricting based on time will unnecessarily penalize studies published recently that are not related to COVID. 

- Complete the first sentence under the "inclusion criteria" sub-heading.

o ["Studies reporting data on the length of any of the three intervals of interest for any cancer site in adult cancer patients presenting with primary cancers….were included"]

- Mention if authors of studies that were potentially eligible but had missing information were contacted—corresponds to Figure 1, Reports Assessed for Eligibility, with exclusion reason Incomplete outcome reporting. If authors were contacted, up to how many times before considered excluded?

o Please correct the spelling error in this section of Figure 1."an interval of interest" rather than "if".

- Complete the first sentence under the subheading "exclusion criteria" similar to comment above. 

- It is unclear what the purpose of the independent and blind screening of 35% of the abstracts retrieved was for. Please clarify if the intention was to pilot and assess concordance between the two reviewers before continuing. Make this explicit in the text as well. 

- Please complete the first sentence under the subheading "data extraction", similar to comments above. 

- Under the "statistical analysis" subheading, please add the number of studies that reported mean intervals (rather than median intervals) in addition to the percentages provided. 

Results:

- Consider including the division of articles from HIC's and LMIC's in the first sentence of the results section. (ex. 300 from HICs and 29 from LMICs, or whatever the true breakdown is). 

- When median intervals are presented, the lower and upper bound must be presented in the body of the text. This comment relates to all sections (abstract, all intervals/results, and the discussion).

- In the fourth paragraph under the subheading "Patient Intervals," please quantify what a large number of studies means by reporting the exact number (see below).

o ["Notable differences supported by a large number of studies included a pooled median of 50 compared to 29 days for breast"]. i.e., is this 50? Report the exact number

- Under the subheading "diagnostic intervals" please quantify few studies by reporting the exact number.

o ["Few studies from less developed countries were available…"] 

- Please report the number of studies representing "small" in the sentence noted below.

o ["72 days, R2=15%, although this difference was much less pronounced and based on only a small number of studies, see Table S7)."]

- Under the subheading "treatment intervals" please report the number of studies implied by "very small"

o ["The number of studies from lower-income countries was again very small, so we discuss the results averaged across all countries (Table 3)."]

- Under the subheading "sensitivity analyses" please quantify what is implied by "in most cases"

o ["In most cases (for example, could be n=7) intervals remained unchanged or changed up to 7 days."]

Discussion: 

- The limitation of the study methods discussed in the second paragraph should be expanded upon and should consider the global context. For example, when grouping leukemias together, CML and ALL to be pooled together, which may have drastically different times to presentation, diagnosis, and treatment. A worked example included in the discussion, like the case of leukemia, would be helpful for all readers and highlight the limitation of this method. 

- In the fourth paragraph, discussing cancer registries, please consider adjusting the language. Cancer registries do exist in lower-income countries, however the infrastructure to support their effectiveness may not be established. Adding "The expansion and creation of cancer registries…." may be more appropriate to recognize the efforts already in place.

o ["The creation of cancer registries or large cancer epidemiological databases in lower-income countries as an investment in national cancer control planning is one of the priorities suggested to reduce cancer care disparities worldwide (14)."]

- The limitation of sample size, such that few studies were available from LMICs, needs to be highlighted, as it is an important and crucial result in and of itself. 

o ["There were fewer differences between high and lower-income countries on the diagnostic and treatment intervals at least partially due to lack of enough data."]

- As a descriptive study, it is important to highlight the implicit result that is the distribution of publications from HICs and LMICs. 

- Please correct the spelling from "polled" to "pooled" in the sentence below.

o ["Our sensitivity analysis showed that the removal of studies with highest potential for bias did not substantially affect polled estimates for the more common cancers"]

Reviewer #2: Alex McConnachie, Statistical Review

Petrova et al have done a systematic review and meta-analysis of studies reporting treatment intervals for (adult) cancer, looking at differences between cancer types and between countries based on national income and the Human Development Index. This review considers the statistical aspects of the paper.

In short, these are very good. The authors use methods derived to combine estimates of medians across studies, which has been shown to work better than previously used methods, based on converting everything to a mean and standard error. My comments are not about the methods, so much as the presentation of the results.

The last two paragraphs of the results report some sensitivity analyses, which generally did not affect the results very much, though for the few exceptions, the authors list some cancers for which treatment intervals are reduced. What these estimates reduced to is reported, but not what from, so without going to find the original estimate in the tables, it is not possible to judge these results. For the benefit of the reader, it would help to give the original and sensitivity estimates in the same place.

Table 1 gives a lot of information, but does not give confidence intervals for the estimated differences between low- and high-income countries. P-values could also be given.

I did not like figure 3. I think forest plots would be better. If low- and high-income estimates are reported in pairs, then the margin of the forest plot could show the estimated difference, with a confidence interval and p-value. Note, I would report the actual p-value, not use stars to indicate significance.

The supplement also needs a little attention, most notably in terms of the table numbering (the tabs of the spreadsheet do not match the contents). In general, I would encourage the authors to present confidence intervals for estimates of differences (can CIs be produced for interval ratios?), and report actual p-values rather than using a system of asterisks.

Reviewer #3: Major comment 1. The text reads: ""This was evaluated using a short form of the 'Aarhus checklist' (4) developed to assess the quality of studies that measure intervals on the cancer treatment pathway. The checklist contains questions regarding interval definitions, measurement, use of theoretical frameworks, discussion of validity, biases, and limitations of measurement, among others. The checklist was completed independently by two reviewers and disagreements were resolved by a third reviewer. Studies with scores <25% were considered high risk and studies with �75% low risk, with the rest considered intermediate (see Supplementary Text 3)." So, the bias assessment process does not seem to include survivorship bias / patient selection. This is a key issue, as survivors (who are able to be surveyed or interviewed, to provide information about their patient interval from symptom onset to presentation) may have atypical diagnostic intervals to those patients who die shortly after their diagnosis. see schema Keeble S et al. International Journal of Cancer 2014 https://pubmed.ncbi.nlm.nih.gov/24515930/ (which forms part of the citation list), and also Koo M et al, Neoplasia 2017 review paper https://pubmed.ncbi.nlm.nih.gov/29253839/ where the matter is explained schematically. 

Major comment 2. There is no theoretical explanation as to why the review excluded the primary care interval - particularly as several studies do exist that report this. This is likely to be a crucial component of any diagnostic process in most (?all) countries, particularly low-income countries which the review focuses on. I am not sure that these data / studies should be excluded - if they are excluded a strong rationale needs to be provided for their exclusion.

More minor points:

1. Abstract, sentence: "The longest diagnostic intervals were observed for hematological, gynecological, and digestive/gastrointestinal cancers, and the longest treatment intervals for prostate cancer, with few studies available from lower-income countries." It would be more informative if specific cancer sites are called out, as these groups include widely different (with respect to diagnostic timeliness) cancers, eg. myeloma (known to have very long diagnostic intervals) versus leukaemia (know to have very short intervals) are both 'haematological', and similarly ovarian (long intervals) vs endometrial (short intervals) are both 'gynaecological', etc. for other cancers. Please could you state specific cancer sites (as exemplars).

2. I found all the material deposited in Open Science Framework very useful and excellent practice, well done to authors' team.

3. Where text reads: "…..such as stage migration (i.e., later stage at diagnosis)" - I think this needs rephrasing, e.g. to read "later stage at diagnosis _because of stage migration" (my underline, to be removed).

4. What does the following mean: "In the case of HDI, the high vs. lower groups were created using k-means clustering." Please explain.

5. In the exclusion criteria (online appendices) many papers are excluded because of reporting the 'WRONG OUTCOME'. However, these studies did not report any wrong outcome per se, only an outcome not being considered by the review. Please rephrase.

Reviewer #4: The authors are congratulated for their valuable manuscript that is a meaningful addition to the scientific literature and to cancer policy development. The meta-analysis methodology is robust and results are presented well. Consideration can be made to strengthen the Discussion and expand the data synthesis. 

Major considerations:

1. Data synthesis and interpretation: while understanding that these sub-analyses may be outside the scope of the current study, the following considerations can be made given the quality of data:

(a) Information on study design and percent of studies utilizing interviews, medical records or database by country: as governments and international stakeholders evaluate the feasibility of including early diagnosis indicators in national programmes, it would be helpful to present the information from Table S4 into the main text. What percentage of countries obtain this information through databases and/or medical records? Can any comment be made on differences in interval estimates according to data source (ie, are patient interviews reliable)?

(b) Total interval: why only a minority of studies evaluate total interval, is there sufficient data to perform a meta-analysis for total intervals for select cancers? 

(c) Variability among high-income countries: the authors could consider a sub-analysis comparing high-income countries. This could include a comparison between high-income countries that have had dedicated public health programmes in early diagnosis (eg, UK). Data on the availability of early diagnosis programmes can be gathered from databases like WHO and/or OECD. 

(d) At risk population: recognizing that this may not have been collected, did the authors identify population groups who were at higher risk for delays? If so, can a comment be made on this? 

2. Discussion: 

(a) Diagnostic and treatment requirements for select cancer types: why have the authors referenced that such a discussion would be beyond the scope and possibilities of this manuscript? If data are being synthesized by cancer, a brief comment highlighting differences between select cancers (particularly those most represented in this study) is relevant and valuable. 

(b) Screening: the authors state that "another strategy that could contribute to reducing the negative impact of lengthy patient delays in this context is the successful implementation of cancer screening programmes." It should be noted that <10% of cancers are found by screening in HIC; screening programmes are not effective means for early diagnosis, particularly in lower-income countries. Such a statement should be strongly referenced if to be endorsed or can be avoided. 

3. Data visualization: the authors present relevant information on relative contribution through interval ratios. It would be informative to convert this into a figure disaggregated by high-income countries and lower-income countries. This will assist the reader to identify particular bottlenecks. 

Minor consideration: 

1. Classification scheme: the authors use categorization of the National Cancer Institute (reference 25). Can the authors provide feedback on why ICD-10 was not utilized? 

2. "Burden" versus "survival": on pg 11, the authors state that "…longer diagnostic intervals for breast cancer and longer treatment intervals for gynecological cancers in lower-income countries, both of which could be contributing to the higher burden…"; would consider exchanging "lower survival" for "higher burden" to increase precision. 

3. Commentary on longer treatment intervals for genitourinary cancers: the authors present a valid hypothesis, but it would be worthwhile to state that this is a hypothesis for the longer treatment interval unless data are available that the more indolent behaviour explains the longer interval. It could also be the result of concerns about treatment morbidity or stigma. 

4. Intervals in Figure 3 (pg 22): all three figures have the title "Patient Interval"; it is probable that the second and third are "Diagnostic" and "Treatment"

[LINK]

---

## [Decision Letter · Decision Letter 2]

22 Aug 2022

Dear Dr. Petrova,

Thank you very much for re-submitting your manuscript "The patient, diagnostic, and treatment intervals in adult patients with cancer from high and lower-income countries: a systematic review and meta-analysis" (PMEDICINE-D-22-00994R2) for review by PLOS Medicine.

I have discussed the paper with my colleagues and the academic editor and it was also seen again by three reviewers. I am pleased to say that provided the remaining editorial and production issues are dealt with we are planning to accept the paper for publication in the journal.

[LINK]

We look forward to receiving the revised manuscript by Aug 29 2022 11:59PM.   

Sincerely,

Beryne Odeny, 

PLOS Medicine

plosmedicine.org

Requests from Editors:

1. Please use square brackets for in-text reference call outs, e.g., “… [6,7]."

Comments from Reviewers:

Reviewer #1: Dear Authors,

Thank you for the opportunity to review this revised manuscript describing patient, diagnostic, and treatment intervals in adults with cancer. The authors have made significant revisions and expanded the capture of their search strategy to reflect the most up-to-date literature. All suggestions by reviewers were adequality incorporated and justified within the text and supplement. Please be careful with spelling in some places (ex. in the second paragraph of the results, please correct the following sentence: "The majority of articles reported data form high-income countries" Correct "form" to "from"). With these changes, this systematic review and meta-analysis using novel methods will be one of the largest and most meaningful contributions to the existing literature. 

Reviewer #2: Alex McConnachie, Statistical Review

I thank the authors for their consideration of my original points. I am happy with all of their responses.

One typo: on page 11, the patient to treatment interval is abbreviated as PI/DI, when I assume it should be PI/TI.

One question: is there a way to apply weights when using a Wilcoxon Rank Sum test to compare groups of studies and generate confidence intervals? I had a quick look online, and couldn't find anything conclusive, but if there is a way to do it, then it might be slightly better.

Reviewer #3: Thank you to the authors for carefully considering and addressing my comments and questions and further improving their manuscript.

[LINK]

---

## [Editor Report · Decision Letter 3]

15 Sep 2022

Dear Dr Petrova, 

On behalf of my colleagues and the Academic Editor, Amitabh Bipin Suthar, I am pleased to inform you that we have agreed to publish your manuscript "The patient, diagnostic, and treatment intervals in adult patients with cancer from high and lower-income countries: a systematic review and meta-analysis" (PMEDICINE-D-22-00994R3) in PLOS Medicine.

PRESS

Sincerely, 

Beryne Odeny 

PLOS Medicine